EMBO
Molecular Medicine

# VEGF blockade enhances the antitumor effect of BRAF[V600E] inhibition

Valentina Comunanza[1,2], Davide Corà[1,2,3], Francesca Orso[3,4], Francesca Maria Consonni[5], Emanuele Middonti[1,2], Federica Di Nicolantonio[1,2] [iD], Anton Buzdin[6,7], Antonio Sica[5,8], Enzo Medico[1,2] [iD], Dario Sangiolo[1,2], Daniela Taverna[3,4] & Federico Bussolino[1,2,3,*]

## Abstract

The development of resistance remains a major obstacle to long-term disease control in cancer patients treated with targeted therapies. In BRAF-mutant mouse models, we demonstrate that although targeted inhibition of either BRAF or VEGF initially suppresses the growth of BRAF-mutant tumors, combined inhibition of both pathways results in apoptosis, long-lasting tumor responses, reduction in lung colonization, and delayed onset of acquired resistance to the BRAF inhibitor PLX4720. As well as inducing tumor vascular normalization and ameliorating hypoxia, this approach induces remodeling of the extracellular matrix, infiltration of macrophages with an M1-like phenotype, and reduction in cancer-associated fibroblasts. At the molecular level, this therapeutic regimen results in a *de novo* transcriptional signature, which sustains and explains the observed efficacy with regard to cancer progression. Collectively, our findings offer new biological rationales for the management of clinical resistance to BRAF inhibitors based on the combination between BRAF[V600E] inhibitors with anti-angiogenic regimens.

**Keywords** angiogenesis; drug resistance; extracellular matrix; myeloid infiltration; vascular normalization
**Subject Categories** Cancer; Skin; Vascular Biology & Angiogenesis

## Introduction

Activating mutations in the BRAF oncogene occur in approximately 7% of human malignancies, including 50–60% of melanomas and 5–8% of colorectal cancers (CRCs) (Davies *et al*, 2002). BRAF mutations are associated with adverse clinical outcomes in melanoma, thyroid carcinoma, non-small-cell lung cancer, and CRC (Cantwell-Dorris *et al*, 2011). The most frequent BRAF mutation (V600E) (Halilovic & Solit, 2008) affects the kinase domain, leading to constitutive activation of the protein. Oncogenic BRAF induces downstream phosphorylation of MEK and ERK, which in turn triggers cell autonomous proliferation even in the absence of extracellular growth factors (Davies *et al*, 2002).

Specific inhibitors of BRAF[V600E], including vemurafenib and dabrafenib, have been approved for treating BRAF-mutant metastatic melanomas (Bollag *et al*, 2010; Flaherty *et al*, 2012; Hauschild *et al*, 2012). Although these drugs show remarkable clinical efficacy and improve overall survival, almost all patients develop resistance and subsequently relapse (Bollag *et al*, 2010; Flaherty *et al*, 2012). A combination of BRAF and MEK inhibition further improves time to progression and overall survival in patients with metastatic melanomas when compared with single-agent BRAF inhibition (Paraiso *et al*, 2010). It is likely that the limited efficacy of combinations targeting a single oncogenic pathway is due to the plasticity and ability of cancer cells to circumvent such a blockade.

Additional strategies must be exploited to increase the efficacy of BRAF[V600E] inhibitors and circumvent or delay the onset of resistance. Several studies on the effects of BRAF[V600E] inhibitors on the cellular transcriptional landscape have envisaged new, attractive preclinical combinations with molecules affecting specific processes, including glucose metabolism (Parmenter *et al*, 2014), the immune response (Hu-Lieskovan *et al*, 2014), and autophagy (Goodall *et al*, 2014), to improve the antitumor effects of BRAF[V600E] inhibitors.

In physiological angiogenesis, the effects of pro-angiogenic molecules are counterbalanced by those of endogenous inhibitors. During tumor angiogenesis, this balance is tipped in favor of new vessel formation. However, the resulting vessels are highly abnormal both structurally and functionally. This balance could be restored by removing the excess of VEGF-A (VEGF) or by blocking

1    Department of Oncology, University of Torino, Candiolo, Italy
2    Candiolo Cancer Institute IRCCS, Candiolo, Italy
3    Center for Molecular Systems Biology, University of Torino, Orbassano, Italy
4    Molecular Biotechnology Center (MBC), Department of Molecular Biotechnology and Health Sciences, University of Torino, Torino, Italy
5    Humanitas Clinical and Research Center, Rozzano, Italy
6    Laboratory of Bioinformatics, D. Rogachyov Federal Research Center of Pediatric Hematology, Oncology and Immunology, Moscow, Russia
7    National Research Centre "Kurchatov Institute", Centre for Convergence of Nano-, Bio-, Information and Cognitive Sciences and Technologies, Moscow, Russia
8    Department of Pharmaceutical Sciences, Università del Piemonte Orientale "Amedeo Avogadro", Novara, Italy
    *Corresponding author. Tel: +39 011 9933347; E-mail: federico.bussolino@unito.it

VEGF signaling, which would induce pruning of abnormal vessels, resulting in vascular normalization characterized by improved perfusion and alleviation of hypoxia. Hence, strategies that favor vascular normalization could improve the efficacy of cancer therapies (Jain, 2014).

We and others have previously demonstrated that BRAF$^{V600E}$ triggers an angiogenic response by modifying the expression profile of angiogenic inducers (Durante *et al*, 2011; Bottos *et al*, 2012; Sadow *et al*, 2014). Thus, enhanced angiogenesis and tumor–stroma cross-talk may represent an additional therapeutic target in the context of BRAF$^{V600E}$-driven tumors. We reported that BRAF$^{V600E}$ inhibition by the vemurafenib analog PLX4720 in xenografts did not reduce the number of tumor capillaries but instead favored vascular normalization (Bottos *et al*, 2012).

Based on this premise, we hypothesized that BRAF-targeted inhibitors could cooperate with anti-angiogenic regimens in the treatment of BRAF-mutant tumors. Specifically, in this work, we investigated the combined effects of PLX4720 and bevacizumab, an anti-VEGF humanized monoclonal antibody, in xenograft models of melanoma and CRC. We report that this dual treatment induces a new genetic program that regulates myeloid cell recruitment and extracellular matrix remodeling and is more efficient than either single agent for controlling tumor growth and the onset of resistance.

## Results

### Dual BRAF$^{V600E}$ and VEGF targeting provides a combinatorial benefit against BRAF$^{V600E}$ mutants tumor growth *in vivo*

Cohorts of CD1-immunocompromised mice bearing A375 BRAF$^{V600E}$-mutant melanoma xenografts were treated with PLX4720, bevacizumab, or a combination of both (COMBO). PLX4720 or bevacizumab alone caused a clear delay in tumor growth, resulting in a 45% or 56% reduction in tumor volume, respectively, compared with vehicle alone. However, neither single treatment induced a regression of initial tumor size. Concurrent administration of the two drugs at the same doses increased antitumor activity, with 88% reduction in tumor volume compared to vehicle and shrinking by 58% the initial tumor size (Fig 1A). Similar results were obtained by treating CRC COLO205 xenografts. PLX4720, bevacizumab, and COMBO, respectively, inhibited of 61,

57, and 80% tumor growth (Fig 1B). These data indicate that bevacizumab improves the efficacy of the therapeutic inhibition of the oncogenic driver BRAF$^{V600E}$.

Then, we evaluated the effect of PLX4720, bevacizumab, and COMBO on lung colonization of the MC-1 cell line, which is an highly metastatic variant of A375 cells (Orso *et al*, 2016) (Fig 1C). Treatments were started 12 weeks after i.v. cell injection and maintained up to week 14. Lung staining by hematoxylin–eosin (HE) revealed that only COMBO treatment significantly reduced lung nodules area compared to controls (vehicle, 5.1 ± 1.4%; COMBO, 0.6 ± 0.1%).

To investigate the mechanism sustaining the observed tumor shrinkage, we measured cellular proliferation and apoptosis in A375 tumors. The number of proliferating cells did not differ significantly among the three treatment groups. Bevacizumab, PLX4720, and COMBO decreased Ki67 staining by 69, 55, and 58%, respectively, compared to controls ($n = 4$ tumors; Fig 1D). By contrast, only COMBO treatment induced a significant increase in apoptosis (22.4 ± 3.3%, $n = 3$ tumors) compared with vehicle (5.7 ± 1.2%, $n = 3$ tumors) as revealed by TUNEL staining; Fig 1E) and by the accumulation of caspase-cleaved cytokeratin 18 fragment (Appendix Fig S1). These findings support that the antitumor activity observed with COMBO treatment depends on an apoptotic program rather than modification of tumor cell cycle.

### Dual BRAF$^{V600E}$ and VEGF targeting promotes vascular normalization and improves tissue perfusion in BRAFV600E xenografts

We analyzed the vascular effects of bevacizumab and PLX4720 on A375 and COLO205 xenografts, by evaluating microvessel density (MVD) and microvascular area (MVA). In A375 tumors, MVD was reduced by 51% after bevacizumab treatment compared with the control (capillaries/mm$^2$: control, 87.9 ± 3.5; bevacizumab, 42.7 ± 2.1). PLX4720 alone or associated with bevacizumab slightly reduced MVD ($n = 5$ tumors; Fig 2A). The treatment with PLX4720, bevacizumab, or COMBO resulted in a similar reduction in MVA compared with the control group (vehicle, 11398.3 ± 497.0 μm$^2$; PLX4720, 7186.7 ± 522.3 μm$^2$; bevacizumab, 5472.4 ± 423.5 μm$^2$; COMBO 6881.1 ± 615.8 μm$^2$) ($n = 6$ tumors; Fig 2A).

The analysis of MVD and MVA in COLO205 xenografts (Appendix Fig S2A) showed a different effect, characterized by increased values of MVD and MVA following PLX4720 treatment,

---

**Figure 1.   Effect of the combination of PLX4720 and bevacizumab on the growth of A375, COLO205, and MC-1 cells harboring BRAF$^{V600E}$ transplanted in athymic nude mice.**

A   Mice bearing established A375 were treated with vehicle ($n = 7$), PLX4720 ($n = 5$); bevacizumab ($n = 7$) or COMBO ($n = 5$). Tumor growth is expressed as % change of the initial tumor. **$P < 0.01$, ***$P < 0.001$ versus vehicle (PLX4720 $P = 0.0049$; bevacizumab $P = 0.0003$; COMBO $P = 4.86E-06$), $^†P = 0.015$ compared to PLX4720.

B   Mice bearing established COLO205 were treated with vehicle ($n = 7$), PLX4720 ($n = 8$); bevacizumab ($n = 7$) or COMBO ($n = 8$). Tumor growth is expressed as % change of the initial tumor. ***$P < 0.001$ versus vehicle (PLX4720 $P = 2.75E-05$; bevacizumab $P = 2.96E-05$; COMBO $P = 2.13E-07$), $^†P = 0.039$ compared to PLX4720.

C   Melanoma MC-1 cells ($5 \times 10^5$) were injected into the tail vein of CD1 mice. Lung colonization was assayed by HE staining and calculating the number of nodules and their total area normalized per the total area of the lungs. The mice analyzed were as follows: start point, $n = 6$; vehicle, $n = 7$; PLX4720, $n = 6$; bevacizumab, $n = 6$; COMBO, $n = 7$. *$P = 0.031$ versus vehicle.

D   Representative images of tumor cell proliferation determined by immunofluorescence Ki67 staining in A375 xenografts treated as indicated. Bar graphs indicate the Ki67$^+$ area/tumor area ($n = 4$ tumors). ***$P < 0.001$ versus vehicle (PLX4720 $P = 2.83E-08$; bevacizumab $P = 4.25E-13$; COMBO $P = 6.16E-08$).

E   Representative images of tumor cell apoptosis determined by immunofluorescence staining with TUNEL in A375 xenografts treated as indicated. Bar graphs indicate the TUNEL$^+$ area/tumor area ($n = 3$ tumors). ***$P = 2.07E-10$ versus vehicle.

Data information: The scale bars represent 1 cm (A, B), 100 μm (D), and 50 μm (E). The results are given as the mean ± SEM. Significance was assessed by one-way ANOVA test followed by *post hoc* pairwise analysis test (A–E).

confirming previous observations (Bottos *et al*, 2012). On the contrary, bevacizumab and COMBO reduced both parameters in this model.

Then, we analyzed the vessel lumen diameter in both models. In A375 xenografts, the average vessel lumen diameter decreased from $52.7 \pm 3.6 \ \mu m$ in vehicle-treated tumors to $35.3 \pm 3.7 \ \mu m$ and

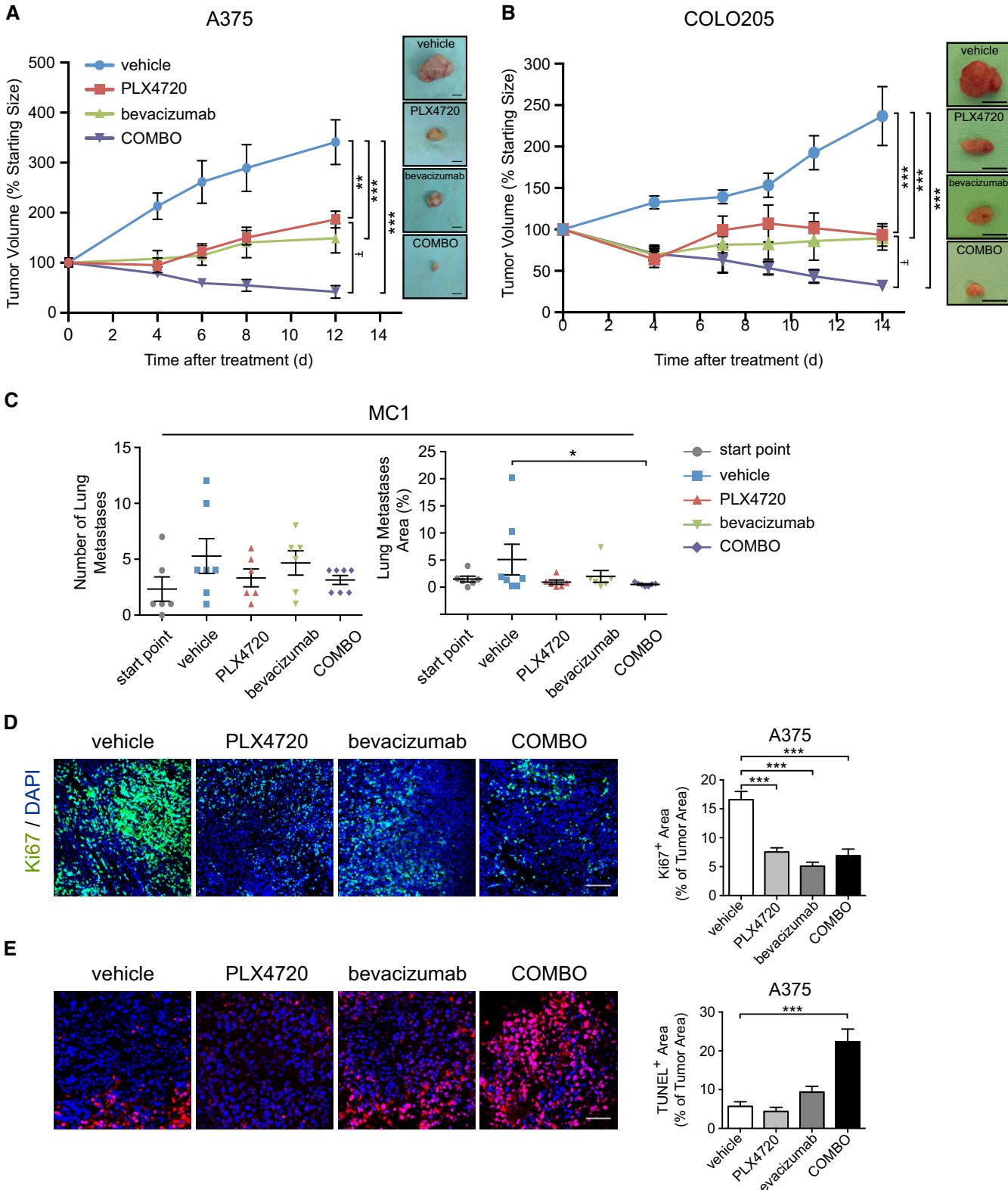

**Figure 1.**

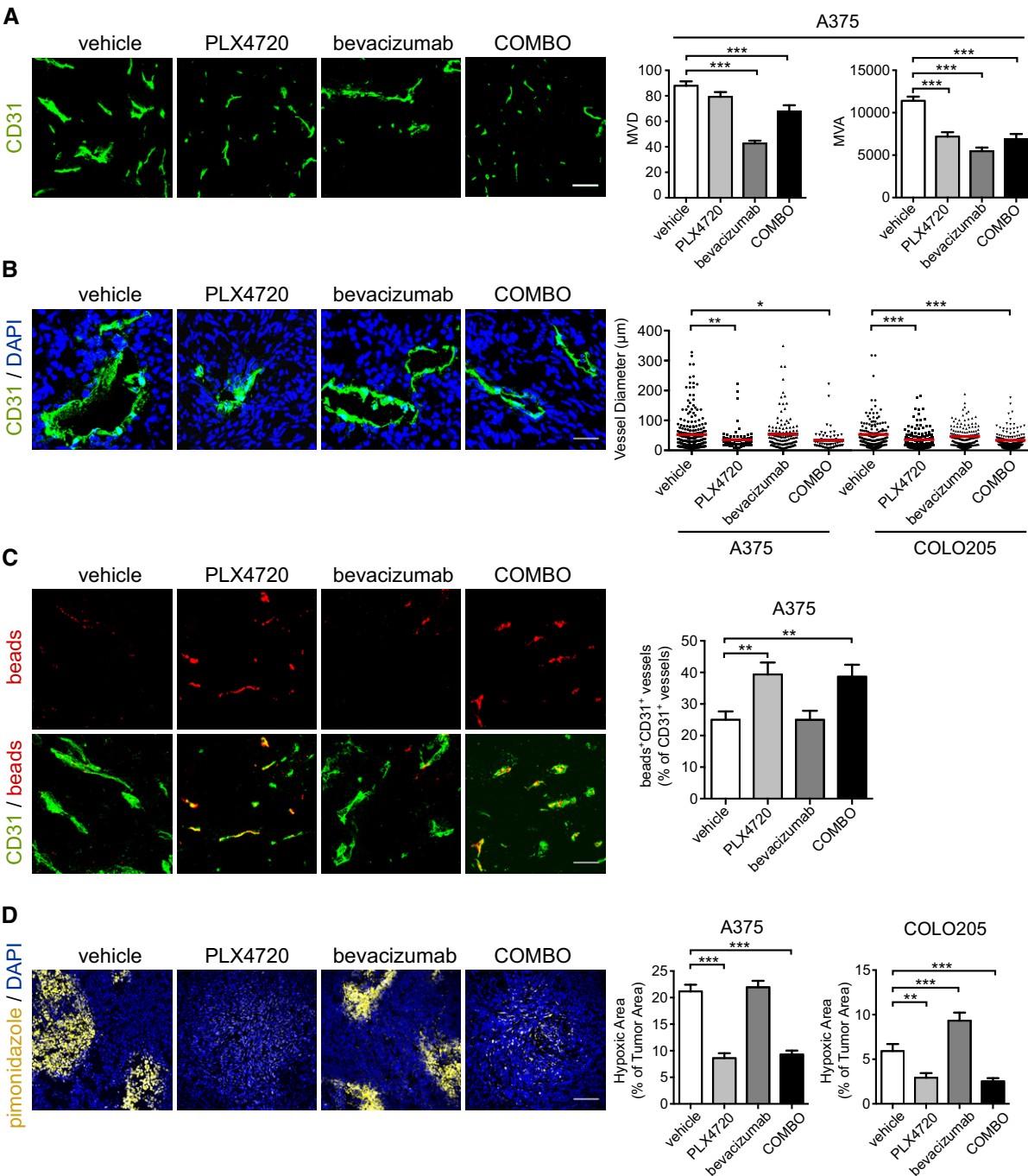

**Figure 2. Vascular response of A375 and COLO205 xenograft tumors to PLX4720 and bevacizumab.**

A   Representative images of vasculature stained by an anti-CD31 antibody in A375 xenografts treated as indicated. Bar graphs indicate quantitative microvessel density (MVD) and microvessel area (MVA) analysis ($n$ = 5 tumors). ***$P$ < 0.001 versus vehicle (MVD: bevacizumab $P$ = 5.94E-25; COMBO $P$ = 1.84E-04) (MVA: PLX4720 $P$ = 4.98E-09; bevacizumab $P$ = 1.16E-19; COMBO $P$ = 3.00E-08).

B   Representative images of vessel lumen in A375 xenografts treated as indicated. Bar graphs indicate the quantitative analysis of lumen diameters in A375 and COLO205 xenografts ($n$ = 3 tumors). *$P$ < 0.05, **$P$ < 0.01, ***$P$ < 0.001 versus vehicle (A375: PLX4720 $P$ = 0.0078; COMBO $P$ = 0.011) (COLO205: PLX4720 $P$ = 2.01E-06; COMBO $P$ = 1.33E-07).

C   Representative images of perfusing fluorescent beads and their relationship with microvessels in A375 xenografts treated as indicated. Bar graphs indicate the % of CD31[+] vessels co-stained with fluorescent beads ($n$ = 4 tumors). **$P$ < 0.01 versus vehicle (PLX4720 $P$ = 0.0031; COMBO $P$ = 0.0033).

D   Representative images of hypoxia marker pimonidazole in A375 xenografts treated as indicated. Bar graphs show the % of tumor hypoxic area in A375 and COLO205 xenografts ($n$ = 5 tumors). **$P$ < 0.01, ***$P$ < 0.001 versus vehicle (A375: PLX4720 $P$ = 6.94E-12; COMBO $P$ = 3.34E-11) (COLO205: PLX4720 $P$ = 0.0015; bevacizumab $P$ = 0.0004; COMBO $P$ = 0.0003).

Data information: The scale bars represent 100 μm in (A, D) and 50 μm in (B, C). The results are given as the mean ± SEM. Significance was assessed by one-way ANOVA test followed by *post hoc* pairwise analysis test.

$33.6 \pm 4.8$ μm upon treatment with PLX4720 and COMBO, respectively ($n = 3$ tumors; Fig 2B). Similar results were observed in COLO205 (vehicle, $50.3 \pm 3.2$ μm; PLX4720, $35.5 \pm 2.3$ μm; COMBO, $33.6 \pm 1.8$ μm). Bevacizumab did not modify the lumen area compared with vehicle in both tumor types.

In spite of the cell line-specific differences (Bottos *et al*, 2012), these data indicate that COMBO and PLX4720 influence vascular morphology by reducing vessel lumen areas as compared to bevacizumab and vehicle. This hypothesis was further investigated in the A375 xenografts. The vessel lumen was resolved and categorized into different groups (from $< 100$ μm$^2$ to $> 600$ μm$^2$). PLX4720 alone or in combination with bevacizumab increased the number of the smallest capillaries (surface $< 100$ μm$^2$) and reduced the number of larger vessels (surface $> 500$ μm$^2$; Appendix Fig S2B). None of the different therapeutic treatments modified the number of vessels with surface areas from 100 to 500 μm$^2$.

These data indicate that BRAF inhibition and VEGF removal have distinct effects on vascular size and that the reduction in number of tumor capillaries induced by VEGF withdrawal is counteracted by BRAF$^{V600E}$ inhibition, which influences capillary shape.

Modification of the vascular area is a component of vascular normalization upon treatment with anti-angiogenic compounds (Jain, 2014). A specific hallmark of this process is the improvement of vascular perfusion, which was studied by the i.v. delivery of orange fluorescent microspheres. PLX4720 or COMBO increased vascular perfusion in A375 xenografts compared with treatment with vehicle or bevacizumab (Fig 2C). As expected, PLX4720 treatment restored tissue oxygenation in this model (Fig 2D) as well in COLO205 xenografts in agreement with our earlier report (Bottos *et al*, 2012). Interestingly, this positive effect was maintained when bevacizumab was combined with PLX4720. Since normalization process may include improvement of pericyte coverage, we assessed the presence of pericytes surrounding capillaries using the specific NG2 marker. We observed that none of the treatments modified pericyte coverage compared with untreated tumors in A375 xenografts (Appendix Fig S2C).

Consistent with our previous study (Bottos *et al*, 2012), we confirmed that PLX4720-mediated BRAF$^{V600E}$ inhibition changed the architecture and functionality of vessels and abrogated tumor hypoxia. Although bevacizumab reduced blood perfusion, its association with PLX4720 did not.

Furthermore, the lack of any evident combinatorial effects between bevacizumab and PLX4720 on tumor vessels, and on the amount of murine VEGF detected in the xenografts (not shown), suggests that the enhanced antitumor activity observed with COMBO is likely independent of angiogenesis.

## Gene expression profiling highlights extracellular matrix remodeling and immunomodulation after dual BRAF$^{V600E}$ and VEGF inhibition

In A375 xenografts, the stromal microenvironment is contributed by the mouse. Therefore, species-specific analysis of gene expression can discriminate the effects of different therapeutic regimens on tumor (human) and stroma (mouse) compartment. Human and mouse microarrays were analyzed separately to determine the dynamic changes within each compartment of the xenograft upon treatment with PLX4720, bevacizumab, or COMBO. Of the 23,025 human genes that were above background, a supervised comparison

(one-way ANOVA with Benjamini–Hochberg (BJ) false discovery rate (FDR) correction) among the four treatment arms revealed a set of 68 differentially modulated genes (Appendix Fig S3A).

Sample clustering analysis based on these 68 genes highlighted minor variations in bevacizumab-treated samples with respect to vehicle and a strong and concordant transcriptional shift in PLX4720- and COMBO-treated samples. A comparison between each treatment arm and the vehicle group by LIMMA (Smyth, 2005) defined a subset of 340 genes modulated by PLX4720 alone (145 up-regulated and 195 down-regulated), 201 genes modulated by COMBO (152 up-regulated and 49 down-regulated), and only four genes modulated by bevacizumab alone (two up-regulated and two down-regulated).

The Venn diagrams in Fig 3A illustrate the overlapping subsets of modulated genes, and the volcano plots (Appendix Fig S3B) show the changes in the log$_2$-fold change and *P*-values for all genes in the three separate comparisons with respect to vehicle. DAVID was used to identify the biological functions enriched in the different treatments in human A375 melanoma cells, considering up- and down-regulated genes separately (significant *P*-value $< 0.05$ after BJ FDR method; Appendix Fig S3C). Genes that were down-regulated in PLX4720 or COMBO groups were highly enriched in gene sets involved in "oxygen levels" and "response to hypoxia". This result is consistent with the observation that tumors are less hypoxic after treatment with PLX4720 or COMBO. Genes that were up-regulated in the PLX4720 and COMBO groups showed enrichment in the Gene Ontology (GO) categories of "immune response", "defense responses", and "inflammatory response". Interestingly, in the GO categories related to immune and inflammatory responses, we identified a cluster of cytokine and chemokine genes. Genes involved in extracellular matrix (ECM) organization, extracellular structure organization, biological adhesion, and cell adhesion were only modulated by treatment with PLX4720 alone (Appendix Tables S1 and S2). Taken together, these analyses indicate highly but not completely overlapping functional processes triggered by PLX4720 and COMBO in the human cell compartment of the tumor. This model does not indicate a significant direct functional signature for bevacizumab treatment on the human cell compartment.

To further characterize the effects of COMBO on the microenvironment of A375 xenografts, we performed mouse-specific gene expression profiling. Of the 14,533 mouse genes that were above background, ANOVA highlighted a larger number of stromal mouse genes that were significantly altered in the four treatment arms compared to the human arrays ($n = 806$, one-way ANOVA after BJ FDR correction). In this case, sample cluster analysis highlighted expression similarities between the vehicle and bevacizumab groups and between the PLX4720 and COMBO groups, with PLX4720 and COMBO inducing a substantially wider gene modulation (Appendix Fig S4A). A subset of bevacizumab-modulated genes related to angiogenesis (compared with vehicle, fold change threshold | log2 FC | of 0.5) is reported and further validated by quantitative PCR analysis (Appendix Fig S4B and C). Volcano plots show the changes in the log$_2$-fold change and *P*-values for all genes for the different treatments (Appendix Fig S4D), and Venn diagrams illustrate the overlapping subsets of modulated genes (Fig 3B). Among the 805 genes modulated by PLX4720 identified by LIMMA analysis, 517 were down-regulated, and 288 were up-regulated. COMBO had a reduced modulatory effect (414 genes; 214 suppressed and 200

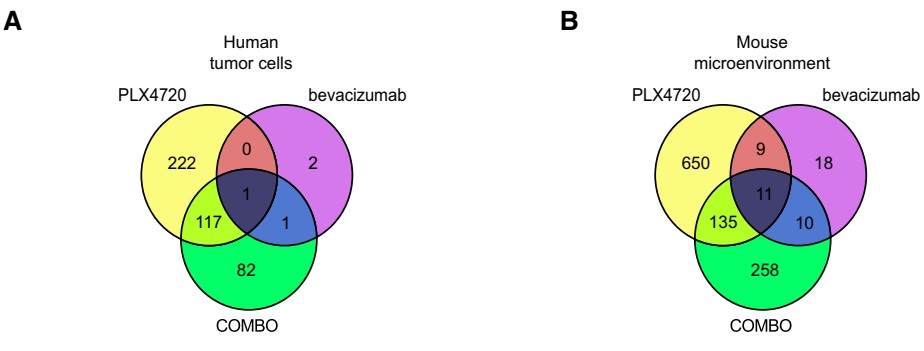

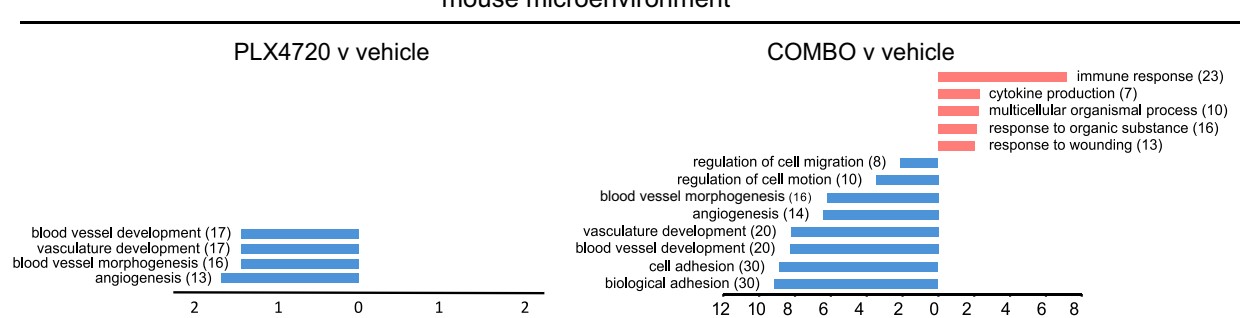

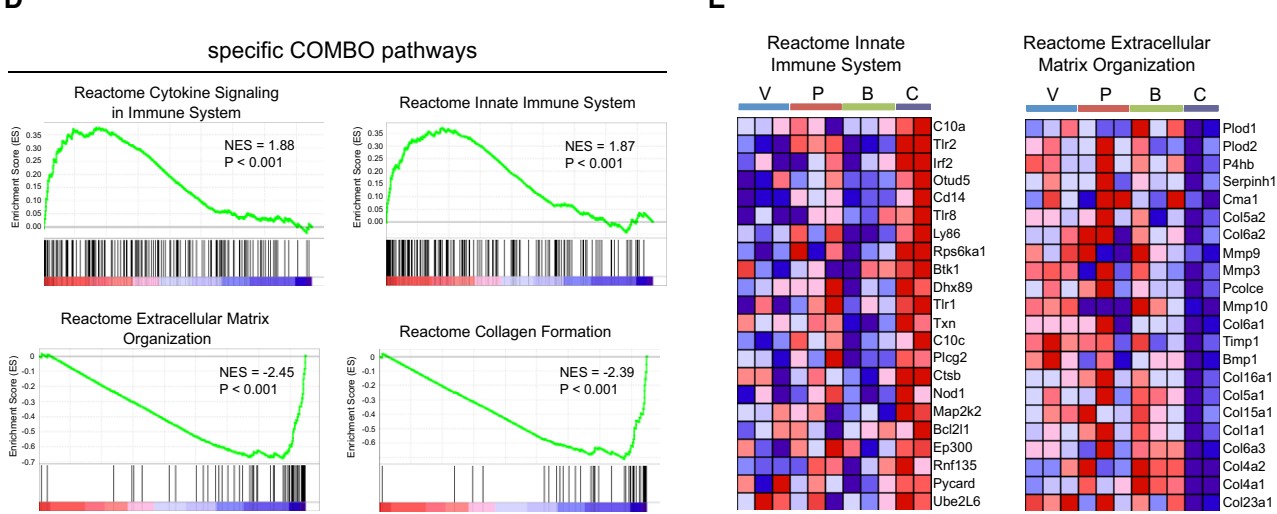

**Figure 3.  Transcriptional profiling in A375 xenografts in response to PLX4720, bevacizumab, and COMBO.**

A   Venn diagram showing the overlapping subsets of modulated human genes (fold change |log₂| ≥ 1, *P*-value < 0.01) in the PLX4720, bevacizumab, and COMBO treatments versus vehicle.

B   Venn diagram showing the overlapping subsets of modulated murine genes (fold change |log₂| ≥ 1, *P*-value < 0.01) in the PLX4720, bevacizumab, and COMBO treatments versus vehicle.

C   Summary of the functional categories of mouse genes significantly enriched in response to PLX4720 and COMBO. GO analyses were performed individually on down- or up-regulated genes using DAVID tool (biological process). GO terms are ranked by *P*-value corrected by BH method, and the number of genes is indicated. For a complete list of significantly enriched GO groups see Appendix Tables S3 and S4.

D   GSEA enrichment plots for "Reactome Cytokine Signaling in Immune System" and "Reactome Innate Immune System" (upper panels) and "Reactome Extracellular Matrix Organization" and "Reactome Collagen Formation" (lower panels) highlight significant enrichment of the pathways relative to the immune response and a decreased expression of the pathways relative to extracellular matrix remodeling in COMBO-treated tumors as compared to the other treatments (vehicle, PLX4720, bevacizumab).

E   Heatmap representation of gene expression changes within the "Reactome Innate Immune System" (left panel) and "Reactome Extracellular Matrix Organization" (right panel) gene set. Genes in heatmaps are shown in rows, and samples are shown in columns. Expression level is represented as a gradient from high (red) to low (blue). V, P, B and C, respectively, indicate vehicle, PLX4720, bevacizumab, and COMBO.

induced). GO analysis was performed on the three treatment arms compared with vehicle using DAVID ($P < 0.05$ with BH FDR correction), considering up- and down-regulated genes separately. Among the down-regulated genes, PLX4720 treatment mainly targeted genes belonging to GO terms characterizing functions and processes of the vasculature ("blood vessel development", "vasculature development", "blood vessel morphogenesis", "angiogenesis"; Fig 3C and Appendix Tables S3 and S4) in accordance with the observed effect on tumor vessel shape and functions (Fig 2). DAVID analysis of the COMBO response revealed that the top enriched functions belonged to the immune response (up-modulated genes), blood vessel, and cell adhesion (down-modulated genes; Fig 3C). A comparison between the lists of genes modulated in A375 xenografts by PLX4720 or COMBO with respect to vehicle revealed a subset of commonly modulated genes and sizeable subsets of genes specifically modulated by one of the two treatments only.

Gene set enrichment analysis (GSEA) was carried out on mouse microarray data set to identify associated biological processes and pathways related to COMBO treatment. By comparing COMBO group with all the other groups, GSEA revealed an enrichment of genes involved in immune response ("Reactome Cytokine Signaling in Immune System" and "Reactome Innate Immune System") and down-modulation of genes associated with the ECM remodeling ("Reactome Extracellular Matrix Organization" and "Reactome Collagen Formation"; Fig 3D). This analysis was confirmed by the heatmaps of genes associated with these categories (Fig 3E and Appendix S4E), indicating that COMBO increased the expression of Toll-like receptors, cytokines and their receptors (Il1β, Il18, Il6r), interferon-related proteins (Irf2), monocytes/macrophage marker (Cd14) and decreased the expression of many collagens. Last, we performed a pathway analysis of the 258 genes that selectively characterized the COMBO treatment, that is, the genes modulated by COMBO versus vehicle. Ingenuity pathways analysis (IPA) revealed gene networks similar to those previously predicted by GO enrichment analysis. In particular, a 24-gene network related to collagen deposition and a 26-gene network clustering around immune cell trafficking displayed extensive regulation (Appendix Fig S4F). Taken together, these results indicate that the combination of PLX4720 with bevacizumab specifically alters alternative biological pathways associated with ECM remodeling, inflammation, and immune response. Interestingly, IPA analyses revealed a network of cytokine and chemokine genes that were also identified in the human transcriptome. TNFα, TGFβ, and IL1β were among the top five IPA upstream regulators that were selectively modulated after COMBO treatment (Appendix Table S5).

## Dual BRAF$^{V600E}$ and VEGF targeting promotes macrophage infiltration with a M1-like phenotype

The evaluation of angiogenic response to COMBO treatment together with the gene expression analysis suggested that vascular changes alone were unlikely to explain the enhancement of tumor inhibition observed after COMBO regimen. Therefore, we investigated whether the immune gene signature promoted by this treatment resulted in recruitment of myeloid cells, which are still present in CD1 athymic mice. Immunofluorescence staining for the pan-hematopoietic cell surface marker CD45 revealed that only COMBO treatment induced marked leukocyte infiltration in A375 xenografts

(CD45$^+$ % area, $24.4 \pm 3.1\%$), which was negligible in vehicle-treated mice ($3.0 \pm 0.5\%$) or after treatment with PLX4720 ($5.1 \pm 0.9\%$) or bevacizumab ($6.9 \pm 1.4\%$) ($n = 5$ tumors; Fig 4A) alone. A similar pattern was seen in COLO205 xenografts. Leukocyte infiltration increased twofold after COMBO treatment (CD45$^+$ % area, COMBO: $9.3 \pm 1.2\%$; vehicle $3.8 \pm 0.5\%$), decreased after PLX4720 treatment ($2.3 \pm 0.3\%$), and increased modestly but not significantly after bevacizumab treatment ($4.9 \pm 1.1\%$) ($n = 5$ tumors; Fig 4A). The features of the infiltrate were further investigated by specific immune phenotype analysis. We observed that F4/80$^+$ (Fig 4B) macrophages represented the most abundant leukocyte population when the BRAF$^{V600E}$ and VEGF pathways were simultaneously blocked in both A375 (F4/80$^+$ area, $27.8 \pm 3.3\%$ vs. $8.4 \pm 1.2\%$ in vehicle) and COLO205 (F4/80$^+$ area, $10.6 \pm 1.5\%$ vs. $2.7 \pm 0.8\%$ in vehicle) xenografts. In both models, the single-drug treatments did not modify the amount of infiltrating F4/80$^+$ cells compared with controls. Immunofluorescence results were further confirmed by flow cytometry analysis (FACS) on untreated and COMBO-treated A375 tumors. Fig 4C shows that COMBO treatment induced a higher CD45$^+$ leukocyte recruitment compared with vehicle, which was mainly constituted by F4/80$^+$ tumor-associated macrophages (TAMs) and CD11c$^+$MHCII$^+$ dendritic cells (DC) (Fig 4D).

To characterize the phenotype associated with the recruited macrophages, we performed species-specific real-time quantitative real-time PCR analysis to assess specific mouse transcripts relative to the M1 and M2 polarization, which, respectively, have antitumor and pro-tumor effects (Mantovani & Sica, 2009). Tumors treated with COMBO expressed significantly higher amounts of M1 markers, such as m-Ccl5, m-Cd40, m-Cxcl10, m-Cxcl9, m-Il1b, m-Stat1, and m-Il1b, compared with vehicle-treated tumors. COMBO-treated tumors also expressed reduced levels of the M2 marker m-Arg1 (Fig 5A), while m-Il10, which characterizes M2 polarization, was not affected. Tumors treated with PLX4720 or bevacizumab alone did not exhibit any significant modulation of macrophage-associated genes (Fig 5A). The ability of COMBO treatment to induce a specific TAM polarization was additionally analyzed by immunofluorescence analysis. Fig 5B shows that COMBO treatment specifically increased the recruitment of CCR7$^+$CD68$^+$ M1 cells without any effect on CD206$^+$CD68$^+$ M2 cells. These data were further confirmed by *ex vivo* experiments on tumor cells isolated from vehicle- and COMBO-treated xenografts and *in vitro* stimulated with ionomycin and phorbol myristate acetate (PMA). Fig 5C shows that COMBO regimen primed CD45$^+$F480$^+$ cells to express more iNOS and TNFα, which characterize M1-polarization (Mantovani & Sica, 2009), than vehicle. These observations suggest that TAMs recruited by COMBO have an M1 phenotype, which can explain the superior effect of the dual therapeutic regimen on tumor burden compared with the effect of mono-therapies (Fig 1).

To discriminate between a systemic or tumor site-specific effect of COMBO, we separately assessed the rate of circulating and tumor-recruited inflammatory monocytes (CD45$^+$Cd11b$^+$Ly6C$^{high}$ cells; Shi & Pamer, 2011). COMBO increased this cell population in A375 tumors but not in bloodstream (Fig 5D and E). CD45$^+$CD11b$^+$/Ly6C$^{high}$ tumor-resident monocytes were further studied by the expression of inflammatory markers TNFα and iNOS (Shi & Pamer, 2011) after *in vitro* stimulation with ionomycin and PMA. As shown in Fig 5D, COMBO regimen enhanced the

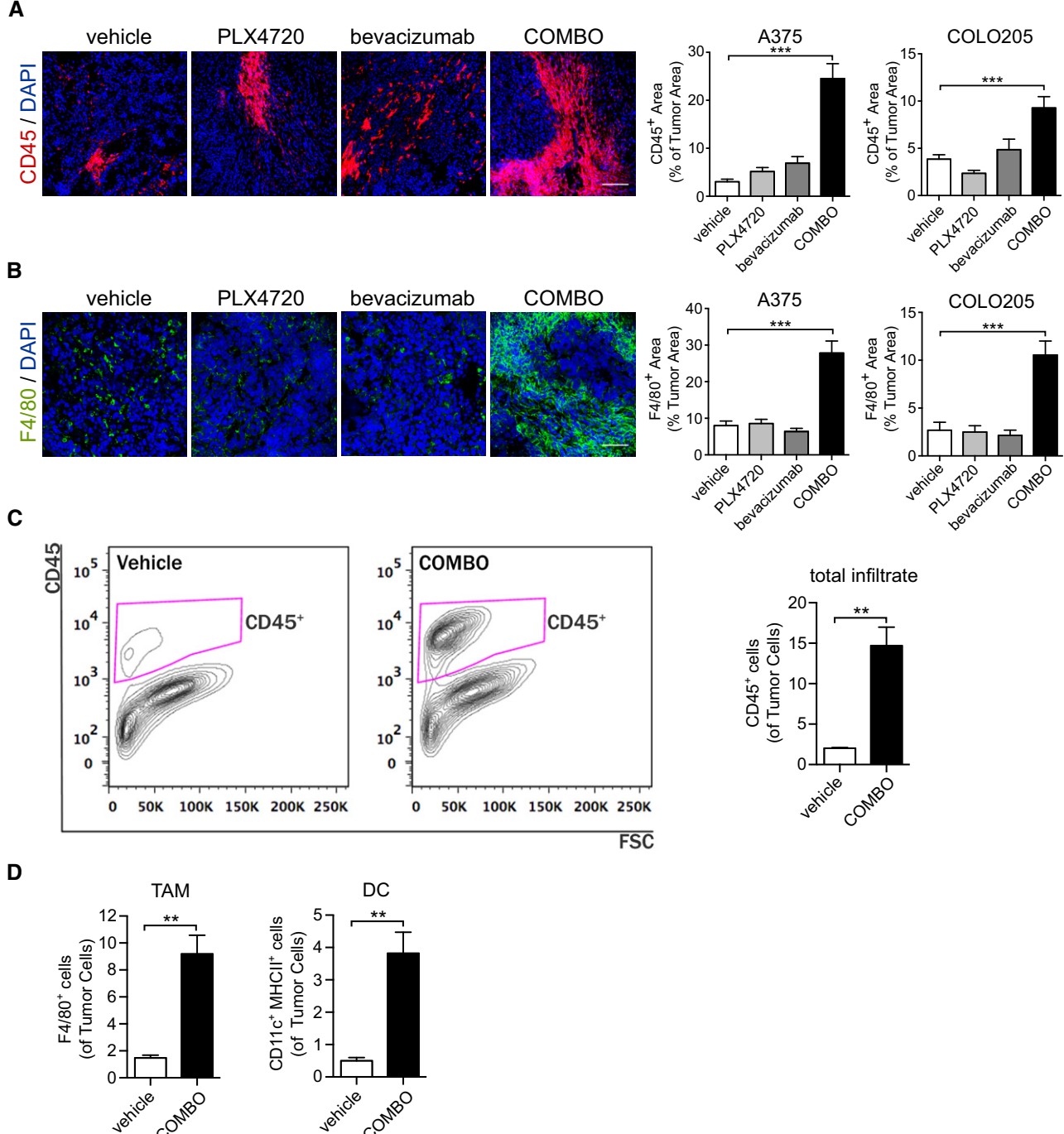

**Figure 4. COMBO treatment increases macrophage infiltration.**

A   Representative images of leukocytes infiltration determined by immunofluorescence CD45 staining of A375 xenografts treated as indicated. Bar graphs indicate the CD45+ area/tumor area in A375 and COLO205 xenografts (*n* = 5 tumors). ***P < 0.001 versus vehicle (A375: COMBO *P* = 2.19E-20) (COLO205: COMBO *P* = 1.81E-06).

B   Representative images of macrophage infiltration determined by F4/80 immunofluorescence staining in A375 xenografts treated as indicated. Bar graphs indicate the F4/80+ area/tumor area in A375 and COLO205 xenografts (*n* = 5 tumors). ***P < 0.001 versus vehicle (A375: COMBO *P* = 3.42E-13) (COLO205: COMBO *P* = 1.94E-06).

C   A375 xenograft tumors after 14 days of treatment were subjected to FACS and tumor infiltrate was analyzed. Gating strategy for CD45+ cells and graph showing the quantification of FACS analysis of infiltrating CD45+ leukocytes in vehicle (*n* = 4 tumors)- compared to COMBO (*n* = 6 tumors)-treated tumors. **P = 0.0064 versus vehicle.

D   Graph showing the quantification of FACS analysis of infiltrating F4/80+ macrophages and CD11c+MHCII+ dendritic cells in vehicle (*n* = 4 tumors)- compared to COMBO (*n* = 6 tumors)-treated tumors. **P < 0.01 versus vehicle (TAM *P* = 0.0061; DC *P* = 0.0091).

Data information: The scale bars represent 100 μm in (A, B). The results are given as the mean ± SEM. Significance was assessed by one-way ANOVA test followed by *post hoc* pairwise analysis test (A, B) and Student's *t*-test (C, D).

**Figure 5.  Macrophages infiltrated after COMBO treatment are polarized toward M1-like phenotype.**

A   Real-time quantitative PCR of the indicated genes (M1-like and M2-like macrophages markers) in A375 xenograft treated with PLX4720, bevacizumab, or COMBO. Data are presented as expression fold change ($\log_2$) compared with vehicle after normalization for housekeeping gene TBP ($n = 3$ tumors). $*P < 0.05$, $**P < 0.01$, $***P < 0.001$ versus vehicle (PLX4720: m-Arg1 $P = 5.65E-06$) (bevacizumab: m-Ccl5 $P = 0.025$) (COMBO: m-Ccl5 $P = 0.0070$; m-Cd40 $P = 0.00099$; m-Cxcl10 $P = 0.043$; m-Cxcl9 $P = 0.046$; m-Il1b $P = 0.0367$; m-Stat1 $P = 0.049$; m-Tlr2 $P = 0.00182$; m-Arg-1 $P = 0.0013$).

B   Representative images of COMBO-treated tumor sections co-stained with an antibody against CD68 and an anti-CCR7 antibody (M1-like polarization) or anti-C206 antibody (M2-like polarization). The scale bars represent 50 μm.

C   Graphs showing the quantification of FACS analysis of infiltrating F4/80$^+$ tumor macrophages and average percentage of median fluorescence intensity (MFI) of TNFα$^+$ or INOS$^+$ on F4/80$^+$ cells after *ex vivo* stimulation with PMA and ionomycin in vehicle ($n = 4$ tumors) and COMBO ($n = 6$ tumors), $*P < 0.05$ versus vehicle (TNFα $P = 0.037$; INOS $P = 0.038$).

D   Graphs showing the quantification of FACS analysis of infiltrating CD11$^+$Ly6C$^+$ tumor monocytes and average percentage of MFI of TNFα$^+$ or INOS$^+$ on CD11$^+$Ly6C$^+$ cells after *ex vivo* stimulation with PMA and ionomycin in vehicle ($n = 4$ tumors) and COMBO ($n = 6$ tumors), $*P < 0.05$ versus vehicle (TNFα $P = 0.048$; INOS $P = 0.038$).

E   Graph showing the quantification of circulating CD11$^+$Ly6C$^+$ monocytes in peripheral blood analyzed by FACS in vehicle ($n = 4$ tumors)- and COMBO ($n = 6$ tumors)-treated tumors.

F   Cell viability measured as fluorescence intensity of ZsGreen-A375 cells co-cultured with cells isolated from xenograft treated with vehicle or COMBO after 48 h. Data are representative of one experiment of two done in triplicate. $*P = 0.046$ versus vehicle.

Data information: The results are given as the mean $\pm$ SEM. Significance was assessed by Student's *t*-test (A–F).

expression of both markers as compared to untreated tumors. Interestingly, when we analyzed the expression of macrophage chemotactic cytokines produced by tumor cells in the xenograft model, we observed a significant increase in human GM-CSF and human TNFα levels after PLX4720 exposure independently from bevacizumab treatment (Appendix Fig S5A). To determine whether this M1-like phenotype correlated with enhanced antitumor effect, we co-cultured the whole cell population isolated from vehicle- or COMBO-treated xenografts with parental Zs-Green-A375 tumor cells to evaluate the cytotoxic effect of leukocytes present in the tumors. Isolated cells from COMBO-treated tumors induced higher cytolytic activity of co-cultured Zs-Green-A375 cells than cells isolated from vehicle tumors (Fig 5F). This result suggests that TAMs recruited by the COMBO regimen display tumoricidal activity, most likely mediated by the M1 phenotype.

COMBO also recruited neuropilin-1 expressing monocytes (NEMs), a novel minute myeloid population with antitumoral and vascular-normalizing effects (Carrer *et al*, 2012). We observed an increase in the CD11b$^+$Nrp1$^+$Gr1$^-$ cells in dual therapy-treated A375 xenografts compared with the controls (NEMs percentage area, $5.2 \pm 0.9\%$ vs. $1.4 \pm 0.3$). Interestingly, PLX4720 alone was ineffective, and bevacizumab showed an inhibitory trend of the basal recruitment rate observed in absence of therapy (PLX4720, $2.2 \pm 0.8\%$, bevacizumab, $0.3 \pm 0.2\%$) ($n = 5$ tumors; Appendix Fig S5B).

The amount of CD11b$^+$Gr1$^+$ cells, which identify myeloid-derived suppressor cells (Shojaei *et al*, 2007), was negligible in untreated tumors and in all treatment conditions (not shown).

**Dual BRAF$^{V600E}$ and VEGF targeting promotes ECM remodeling**

The microarray data also highlighted ECM remodeling in A375 xenografts. We therefore examined whether matrix deposition was also affected by BRAF$^{V600E}$ and VEGF blockade. We observed that COMBO treatment reduced the amount of type I collagen area fraction (COMBO: $7.3 \pm 0.9\%$ vs. vehicle: $14.9 \pm 1.1\%$), while neither single agent was effective (type I collagen-positive area, PLX4720 $14.6 \pm 1.3\%$; bevacizumab: $16.7 \pm 0.9\%$) ($n = 5$ tumors; Fig 6A). The significant reduction in collagen I in COMBO-treated tumors indicates that blockade of BRAF$^{V600E}$ and VEGF induces the degradation of matrix or prevented matrix

production. In tumors, ECM dynamics are largely orchestrated by cancer-associated fibroblasts (CAFs) (Kalluri & Zeisberg, 2006). Therefore, we investigated whether COMBO treatment could also reduce CAF density. Tumor sections were stained with an antibody against αSMA, a specific marker of CAF and pericytes. We observed that COMBO treatment lowered the density of non-perivascular αSMA-positive cells, suggesting a decrease in CAFs ($n = 5$ tumors; Fig 6B).

Of relevance, microarray analyses of murine gene expression data indicated that proteolytic enzymes, including matrix metalloproteinases (Mmp3, Mmp9, Mmp10; Fig 3E) were reduced after COMBO treatment. Therefore, it is plausible that the reduced amount of collagen I in COMBO-treated tumors mainly reflects impaired deposition rather than increased degradation. We investigated also the expression of lysyl oxidase (LOX), which regulates tissue stiffness by collagen cross-linking (Barker *et al*, 2012). Of interest, COMBO treatment reduced the LOX expression (COMBO: $3.5 \pm 0.7\%$ vs. vehicle: $6.6 \pm 0.8\%$), while PLX4720 single-drug treatment did not alter its expression and bevacizumab modestly increased its levels ($n = 3$ tumors; Fig 6C). Accordingly, IPA analyses of murine gene modulated by COMBO indicated that TGFβ1 signaling, which is involved in regulating the cellular production of ECM molecules and CAF differentiation (Shukla *et al*, 2014), was a top upstream regulator of the class of genes negatively modulated by COMBO (Fig 6D and Appendix Table S5). The value of this computational prediction was experimentally confirmed. COMBO did not significantly modify the abundance of murine *Tgfb1* mRNA, but it was the most effective treatment in reducing the level of human TGFβ analyzed by a pan TGFβ antibody and of human TGFB1 transcript (Fig 6E and F).

**TAMs recruited by COMBO regimen are instrumental in the enhanced antitumor effect**

To explore the role of TAMs in the enhanced tumor activity observed in COMBO treatment, clodronate liposomes were used to deplete macrophages during treatments. Clodronate alone promoted a tumor inhibitory effect as previously reported in other tumor models (Fischer *et al*, 2007). Nevertheless, clodronate co-administration impaired the effect of COMBO, which had a cytostatic effect without inducing tumor shrinkage (COMBO plus clodronate:

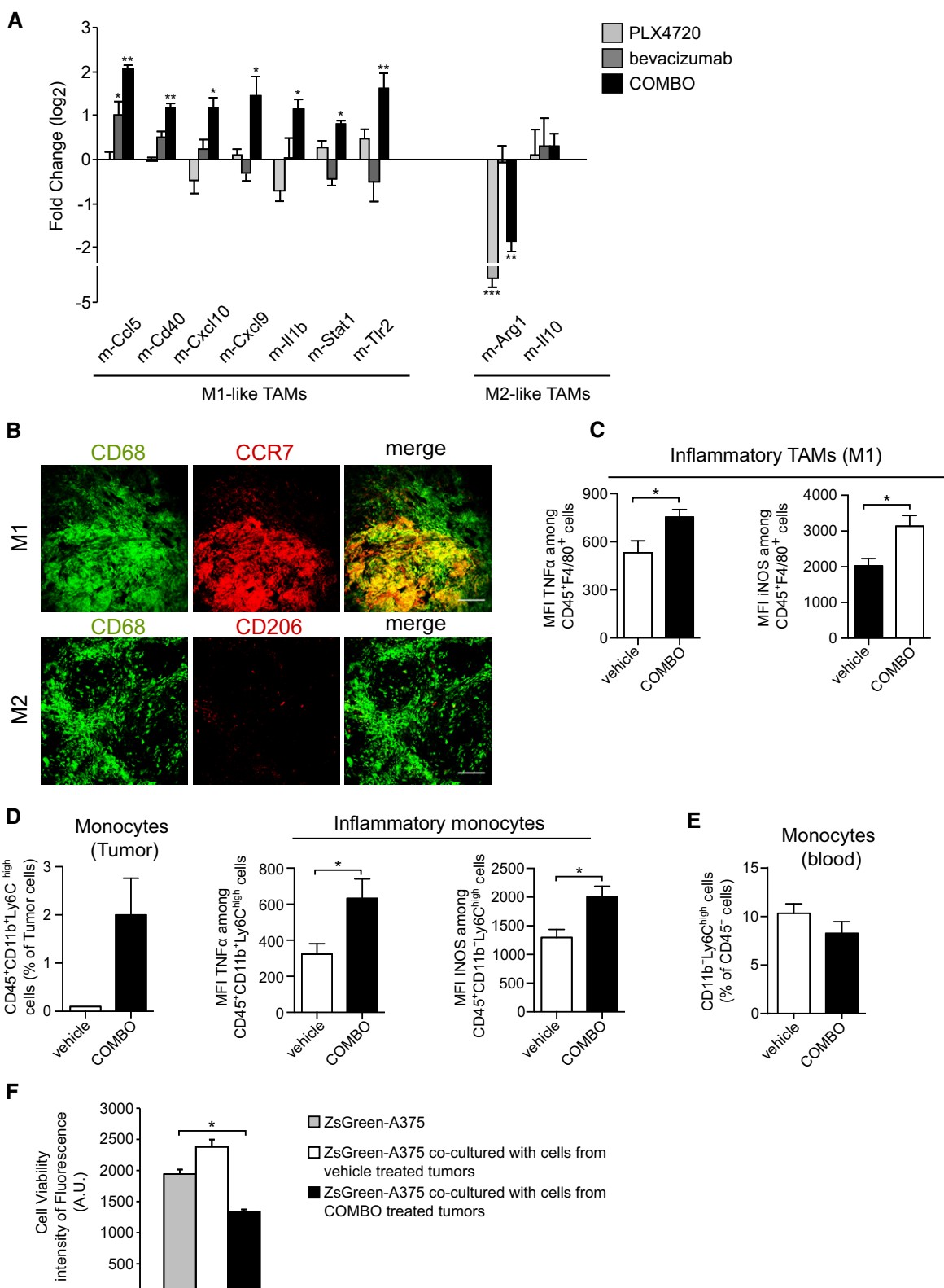

**Figure 5.**

129.6 ± 35.2% of tumor starting size, COMBO: 33.2 ± 14.6% of tumor starting size; Fig 7A). Immunofluorescence staining of F4/80+ cells confirmed that clodronate liposomes really depleted

macrophages in COMBO-treated A375 tumors (F4/80+ area, 23.5 ± 1.7% in COMBO group and 2.2 ± 0.2% in COMBO clodronate group; Fig 7B). Interestingly, the decreased type I collagen

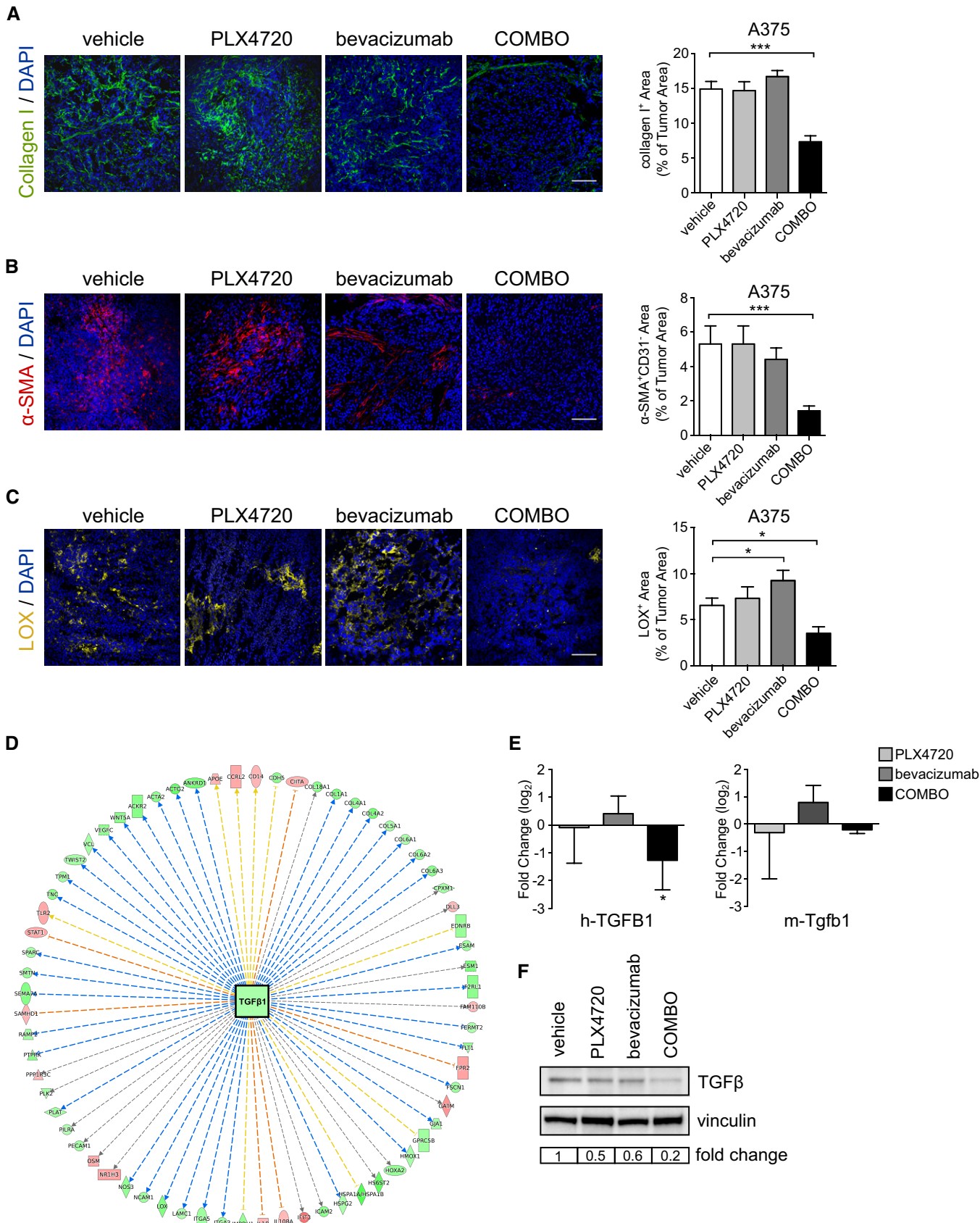

**Figure 6.**

**Figure 6.  COMBO treatment reduces collagen deposition in A375 xenografts.**

A   Representative images of collagen deposition determined by collagen I immunofluorescent staining in A375 xenografts treated as indicated. Bar graphs indicate the collagen I[+] area/tumor area ($n = 5$ tumors). ***$P = 2.93E-06$ versus vehicle.

B   Representative images of CAF density determined by α-SMA immunofluorescence staining in A375 xenografts treated as indicated. Bar graphs indicate α-SMA[+] area/tumor area ($n = 5$ tumors). ***$P = 0.0006$ versus vehicle.

C   Representative images of lysyl oxidase enzyme determined by LOX immunofluorescence staining in A375 xenografts treated as indicated. Bar graphs indicate the LOX[+] area/tumor area ($n = 3$ tumors). *$P < 0.05$ versus vehicle (bevacizumab $P = 0.033$; COMBO $P = 0.040$).

D   Ingenuity pathway analysis (IPA) identified TGFβ1 as one of the most significant upstream regulators (activation $z$-score $= -4.753$, $P = 2.1E10E-29$; predicted activation state: inhibited) of the specifically modulated genes after COMBO treatment, according to microarray analysis. Shades of red indicate the degree of up-regulation while shades of green indicate the degree of down-regulation of significant downstream genes.

E   Quantitative real-time PCR of h-TGFB1 and m-Tgfb1 of A375 xenograft treated with PLX4720, bevacizumab, or COMBO. Data are presented as expression fold change ($\log_2$) compared with vehicle after normalization for the housekeeping gene TBP ($n = 3$ tumors). *$P = 0.043$ versus vehicle.

F   Western blot for human TGFβ in total protein extract from A375 tumors treated as indicated. Vinculin was used as an internal control.

Data information: The scale bars represent 100 μm in (A–C). The results are given as the mean ± SEM. Significance was assessed by one-way ANOVA test followed by *post hoc* pairwise analysis test (A–C) and Student's *t*-test (E).
Source data are available online for this figure.

deposition induced by COMBO treatment still persisted in the presence of clodronate liposomes (Fig 7C), suggesting that the ECM remodeling mostly depended on a direct effect of the combination of VEGF and BRAF inhibition.

### COMBO regimen delays the onset of acquired resistance to BRAF[V600] inhibition by PLX4720

Clinical experience with the PLX4720 analog vemurafenib recently demonstrated that the efficacy of long-term treatment for patients with melanoma is hampered by the inevitable development of acquired resistance to the drug (Bollag *et al*, 2010; Flaherty *et al*, 2012). Therefore, we investigated the therapeutic effect of PLX4720 and bevacizumab alone or in combination over a period of 6 weeks in melanoma A375 xenografts. After 2 weeks of treatment, detectable tumor relapse progressively occurred in mice that received mono-therapy with PLX4720 or bevacizumab. After 6 weeks, tumor volumes were comparable to untreated controls (Appendix Fig S6A). In COMBO-treated mice, only three of six tumors resumed growth after 6 weeks while three were still in response (one complete, two partial; Fig 7D).

Interestingly, infiltrating F4/80[+] macrophages were still massively detectable in the responding mice but almost absent from relapsed ones (F4/80[+] area, 29.8 ± 3.4% in responder

mice and 6.7 ± 0.8% in relapsed mice; Fig 7E). In accordance, relapsing mice showed an increase in type I collagen area (collagen I[+] area, 13.6 ± 1.6% in responder mice and 21.6 ± 1.1% in relapsed mice; Fig 7F). Of note, the transcript of pro-fibrotic cytokine TGFβ1 (Shukla *et al*, 2014) was increased in tumor cells of relapsing mice (Fig 7G). We also confirmed the M1 phenotype of macrophages recruited in responder mice (Appendix Fig S6B). The level of transcripts encoding GM-CSF, which activates and primes leukocytes, was also higher in responder than in relapsing tumors ($n = 3$ tumors; Appendix Fig S6C). Furthermore, as indicated by CD31 staining, relapsing tumors still exhibited MVA reduction (Appendix Fig S6D), which was accompanied by an increase in hypoxic area ($n = 3$ tumors; Appendix Fig S6E).

Collectively, these results indicate that M1-like macrophages recruited by COMBO prolonged remission in mouse xenografts and effectively helped to delay the onset of acquired resistance to PLX4720.

## Discussion

The connections between cancer and stroma cells are a largely unexplored area of cancer biology. These connections can deeply

**Figure 7.  Bevacizumab delays tumor relapse after acquired resistance to PLX4720 in A375 xenografts through macrophage recruitment.**

A   Effect of clodronate liposomes on tumor growth inhibition induced by COMBO treatment in mice bearing A375 tumors. Tumor growth is expressed as % change of the initial tumor. Macrophage depletion by clodronate liposome enhanced the tumor growth of tumors treated with the combination of PLX4720 and bevacizumab ($n = 5$ mice/group). *$P = 0.049$ vehicle + clodronate versus vehicle, ***$P = 5.72E-06$ COMBO versus vehicle, †$P = 0.033$ COMBO + clodronate versus COMBO.

B   Representative images of macrophage infiltration determined by F4/80 immunofluorescence staining in A375 xenografts treated as indicated. Bar graphs indicate the F4/80[+] area/tumor ($n = 3$ tumors), ***$P = 1.53E-25$ versus vehicle.

C   Representative images of collagen deposition determined by collagen I immunofluorescence staining in A375 xenografts treated as indicated. Bar graphs indicate the collagen I[+] area/tumor ($n = 3$ tumors).

D   Waterfall plots showing the percent change in volume (relative to the initial tumor volume) for the individual A375 xenografts in each treatment group (vehicle, PLX4720, bevacizumab, and COMBO) from week 1 to week 6.

E   Representative images of macrophage infiltration determined by F4/80 immunofluorescence staining in a responder and a relapsing A375 xenografts. Bar graphs indicate the F4/80[+] area/tumor ($n = 3$ mice). ***$P = 1.25E-06$ versus vehicle.

F   Representative images of collagen deposition determined by collagen I immunofluorescence staining in a responder and a relapsing A375 xenografts. Bar graphs indicate the collagen I[+] area/tumor ($n = 3$ mice). ***$P = 4.03E-05$ versus vehicle.

G   Real-time quantitative PCR of h-TGFB1 in A375 xenograft treated with PLX4720, bevacizumab, or COMBO. Data are presented as expression fold change ($\log_2$) of relapsing tumors compared to responder tumors after normalization for the housekeeping gene TBP ($n = 3$ tumors). ***$P = 0.0006$ versus vehicle.

Data information: The scale bars represent 100 μm. The results are given as the mean ± SEM. Significance was assessed by one-way ANOVA test followed by *post hoc* pairwise analysis test (A) and Student's *t*-test (B, C, E–G).

influence not only cancer progression but also the clinical responses to oncogene-targeted therapies (Mantovani & Allavena, 2015).

Here, we demonstrate that targeting the vascular compartment of tumor tissue with bevacizumab can modulate response to PLX4720 of melanoma xenografts harboring the BRAF[V600E] mutation. The

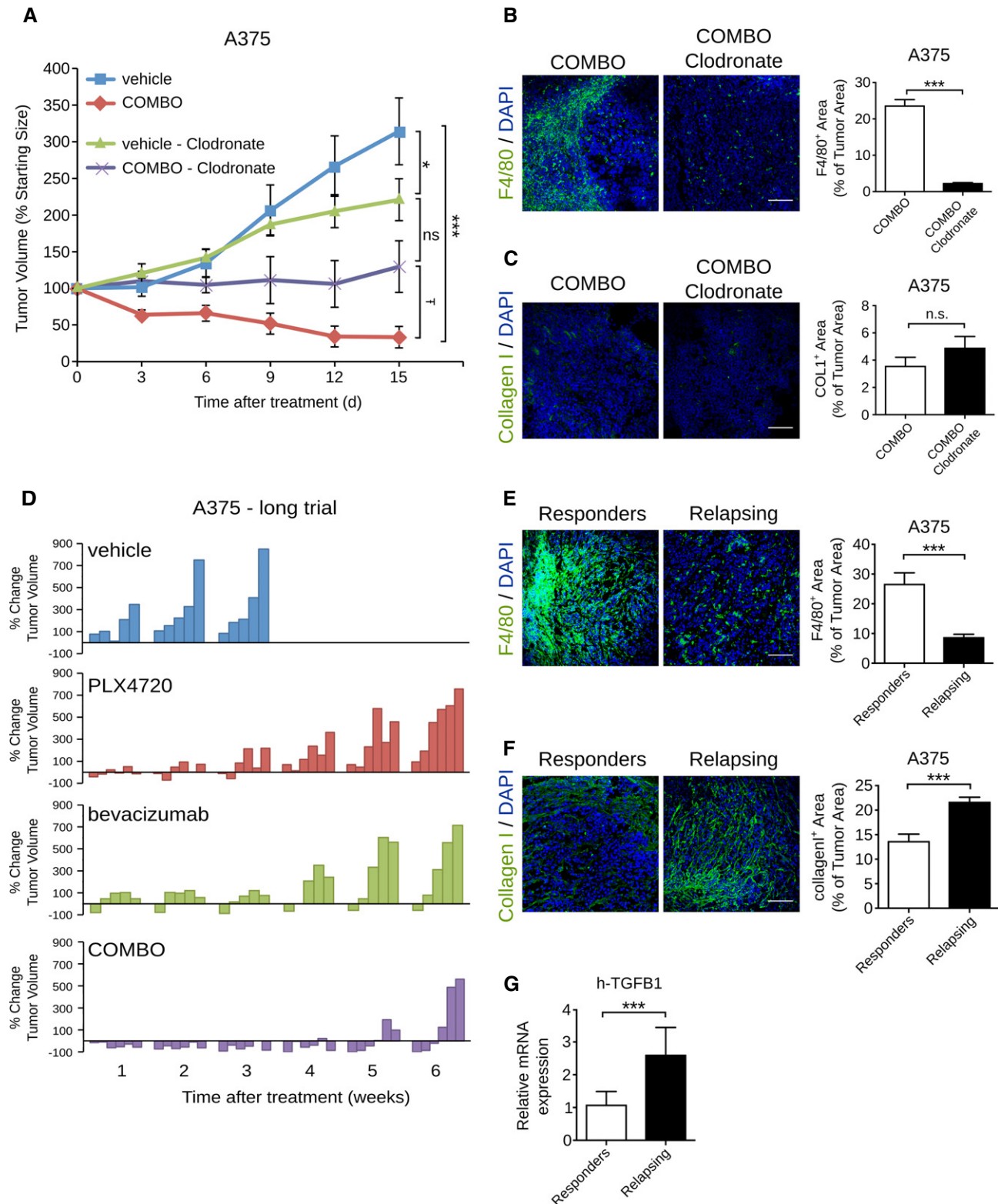

Figure 7.

final result of the drug combination is an enhanced anticancer effect and a delay of the onset of resistance to PLX4720. This effect is the result of the following biological processes, which are not triggered by treatment with either drug alone: (i) the recruitment of M1 polarized TAMs, which have efficient antitumor activity (Mantovani & Sica, 2009); and (ii) remodeling of the ECM characterized by a reduction in collagen I and in CAFs. This cooperative effect is mediated by the adoption of a *de novo* transcriptional signature (neomorphic effect) (Pritchard *et al*, 2013), that was not observed in the presence of either agent alone.

The COMBO treatment triggers a neomorphic footprint mainly characterized by the presence of M1-like TAMs. This immune response is sustained by a general modification of the entire tumor transcriptional landscape characterized by an enrichment of genes related to leukocyte recruitment. We hypothesize that the COMBO promotes the recruitment of inflammatory monocytes that undergo M1 polarization switch necessary for the enhanced efficacy of this regimen on A375 xenografts compared to single agents. This hypothesis is based on the *in vitro* cytotoxic effect on A375 cells of the bulk tumor cell population isolated from COMBO-treated mice, but not from untreated mice had. Furthermore, the comparative analysis of responder and relapsing A375 xenografts after long-term COMBO treatment demonstrated that M1-like TAM infiltrate persists in the former but not in the latter. Interestingly, BRAF[V600E] inhibition by PLX4720 can dampen the immune-suppressive activity observed in melanomas (Khalili *et al*, 2012; Frederick *et al*, 2013).

A second effect of COMBO treatment on myeloid compartment is an increase in a subpopulation expressing neuropilin-1 and CD11b, which have been reported to favor vascular normalization and inhibit tumor growth when directly injected in B16.F10 melanomas at relative high cell concentration ($2 \times 10^5$ cells/tumor) (Carrer *et al*, 2012).

The second neomorphic effect of COMBO influences genes involved in ECM remodeling and cell-matrix adhesion. Similar to the signature reported by Hirata *et al* (2015), PLX4720 monotherapy of A375 xenografts increased the transcription of human genes involved in ECM organization and biological cell adhesion. The administration of PLX4720 in combination with bevacizumab counteracted this signature and reduced the number of CAFs and the amount of collagen I. As reported for M1-like TAM infiltrate, responder mice to the COMBO continue to show reduced amount of collagen, but relapsing do not. These data confirm and extend the relevance of ECM and CAFs in priming resistance to BRAF[V600E] inhibition. Paradoxically, CAFs are activated by vemurafenib or its analogue PLX4720 and they produce hepatocyte growth factor (HGF), which re-activates the MAPK signaling pathway, resulting in BRAF inhibition. This effect is reversed by blocking the HGF/MET axis (Straussman *et al*, 2012; Wilson *et al*, 2012). A second mechanism has been recently identified and demonstrates that the paradoxical activation of fibroblastic stroma triggered by PLX4720 modifies tumor stiffness, which provides an alternative route of MAPK activation by an integrin-mediated outside-in effect, resulting in cancer survival (Hirata *et al*, 2015).

COMBO does not increase the expression of ECM degrading MMPs, but it reduces the expression of LOX, a key enzyme in the formation of collagen fibrils (Barker *et al*, 2012), supporting the notion that VEGF removal by bevacizumab can balance the effect of PLX4720 on ECM and its negative consequence on therapeutic response.

However, other mechanisms could explain the effect of COMBO treatment on the fibroblast compartment. Interestingly, vascular normalization induced by the calcium antagonist verapamil with cilengitide, an integrin mimetic peptide, reduces ECM deposition and the desmoplastic reaction in a model of pancreatic ductal adenocarcinoma (Wong *et al*, 2015).

The effect of COMBO on ECM deposition can also affect the inflammatory response. For example, mice lacking the ECM glycoprotein SPARC have an increased number of macrophages in tumors (Puolakkainen *et al*, 2004). Additionally, collagen I has inhibitory effects on immune cells, attenuating their tumor-suppressive function. Collagen inhibits the ability of macrophages to kill cancer cells by blocking their polarization and activation (Kaplan, 1983). Collagen deposition in tumor microenvironment has also been directly implicated as a barrier to T-cell entry and subsequent cytotoxic activity (Salmon *et al*, 2012). Finally, LOX inhibition in transgenic mice developing pancreatic adenocarcinoma prolongs tumor-free survival and is associated with decreased fibrillar collagen and increased infiltration of macrophages and neutrophils (Miller *et al*, 2015). In general, strategies alleviating the forces that cause vessel compression improve perfusion and subsequent cell trafficking and delivery of drugs (Stylianopoulos & Jain, 2013).

The third neomorphic effect of COMBO concerns the ability of PLX4720 to modulate the effect of bevacizumab on blood vessels. Bevacizumab and, in general, all drugs targeting the VEGF/VEGF receptors axis reduce tumor capillaries by abrogating the pro-survival and proliferative effects of VEGF (Ferrara *et al*, 2003). In principle, drastic shrinkage of tumor vasculature promotes hypoxia, which can favor tumor progression in preclinical models (Ebos *et al*, 2009; Paez-Ribes *et al*, 2009). Based on our previous observations (Bottos *et al*, 2012), PLX4720 normalizes the tumor vasculature with a reduction in hypoxic areas and improves blood flow. These effects also persist in the presence of bevacizumab, which alone does not improve basal perfusion and tumor oxygenation. In accordance with these functional observations, PLX4720 and COMBO treatments result in decreased expression of genes regulating angiogenesis and the response to hypoxia.

The vascular-normalizing effect of COMBO could function cooperatively with the above neomorphic effect on macrophage recruitment and indirectly favor the adaptive immune response. This hypothesis is supported by the evidences accumulated in responder and relapsing mice characterized by the correlation between the normoxic state and the abundance of TAMs. Substantial evidences suggest that the rescue of the hypoxic state by vascular perfusion reprograms the tumor microenvironment from immunosuppressive to immunosupportive. Vascular normalization facilitates the infiltration of cytotoxic T cells (Huang *et al*, 2012), and M2 and M1 polarizations are, respectively, promoted by hypoxia and normoxia (Casazza *et al*, 2013; Laoui *et al*, 2014). Hypoxia also recruits regulatory T-cell (Treg) effectors for tumor tolerance (Facciabene *et al*, 2011), and up-regulates the expression of programmed death (PD-1) ligand 1 (Noman *et al*, 2014), which favors the suppressive activity of MDSC (Greenwald *et al*, 2005). Among the immunosuppressive factors produced by tumor cells, VEGF plays a key role in suppressing dendritic cell maturation (Gabrilovich *et al*, 1996), thus

impairing an effective tumor antigens presentation and the rise of specific antitumor immunity. VEGF promotes MDSC differentiation (Lechner *et al*, 2010) and regulatory T-cell proliferation (Terme *et al*, 2013). Then, FLT-1 expressing monocytes can be recruited by VEGF into the tumor (Mantovani & Sica, 2009) and act together with the Th2 cytokine IL-4 to promote M2 polarization of macrophages (Linde *et al*, 2012). The immune-suppressive role of VEGF was further highlighted by the delayed melanoma growth observed after its silencing. VEGF down-modulation resulted in a decrease in Tregs and MDSC and an increased effector T-cell activation in tumor infiltrates. Together, these events restored tumor sensitivity to treatment with antibodies against the checkpoint inhibitors PD-1 and CTLA-4 (Courau *et al*, 2016).

Our findings provide an intriguing experimental platform to explore possible synergism with immunotherapy strategies and support a previous study performed in mice transplanted with a BRAFV600E melanoma cell line poorly sensitive to PLX4720 (Knight *et al*, 2013). In this model, the BRAF inhibitor did not increase the tumor immunogenicity, but it favored the recruitment of CD8 and natural killer cells. PLX4720 treatment significantly reduces the production of tumoral VEGF, enhancing tumor infiltration by antigen-specific T lymphocytes (Liu *et al*, 2013). In general, contrasting the VEGF effect along with simultaneous BRAF inhibition can turn into a promotion of adaptive immune responses. From a clinical perspective of the treatment of BRAF-mutated melanomas, this synergism could be exploited with either immune-checkpoint modulators (Anti-PD1 Ab or Anti CTLA-4) (Postow *et al*, 2015) or with approaches based on adoptively transferred antitumor lymphocytes (Kwong *et al*, 2013). Antitumor T cells avoid hypoxic areas, and tissue oxygenation restores their activity and efficacy of checkpoint inhibitors (Hatfield *et al*, 2015). Interestingly, bevacizumab shows synergistic effect with ipilimumab, which blocks CTLA4, improving survival in patients with metastatic melanoma (Hodi *et al*, 2014). Therefore, our findings provide biologic strong rational to explore, within dedicated immunocompetent models, the potential beneficial modulation of antitumor adaptive immune response consequent to the double VEGF and BRAF inhibition.

Our holistic experimental approach is poorly instructive on the molecular players involved by COMBO and on the cellular targets. However, bioinformatic predictions experimentally validated (Fig 6) suggest that human TGFβ1 is one of the strong candidate to orchestrate the neomorphic effects of COMBO in A375 xenografts. Actually, the pleiotrophic roles of TGFβ1 include the suppressive activity (Yang *et al*, 2010), the capability to promote CAF differentiation and ECM remodeling (Evans *et al*, 2003; Shukla *et al*, 2014), and the vascular regulatory functions (Jakobsson & van Meeteren, 2013). On the basis of the differential transcriptomic effects of bevacizumab and PLX4720 alone, we can predict that bevacizumab has an exclusive effect on stroma. However, PLX4720 has a wider effect on both tumor and stromal cells, as also reported by the literature (Straussman *et al*, 2012, Wilson *et al*, 2012; Hirata *et al*, 2015).

Overall, our data suggest that in preclinical models the removal of VEGF can increase the efficacy of BRAF^V600E inhibitors by inducing vascular normalization, overcoming immune tolerance and modifying the role played by CAFs. Further experiments will be required to investigate how T and B cells, which are absent in CD1 mice used in this work, modify the functional performance of the association between BRAFV600E targeting and VEGF removal.

# Materials and Methods

### Cell lines and reagents

MC-1 cells were kindly provided by L. Xu and R.O. Hynes (Massachusetts Institute of Technology, Boston) and maintained in DMEM containing Glutamax™ (GIBCO Invitrogen Life Technologies, Carlsbad, CA, USA) and supplemented with 10% fetal bovine serum (FBS), 1 mM sodium pyruvate, 25 mM HEPES pH 7.4, 1 × MEM vitamin solution, 1 × MEM non-essential amino acids and 100 μg/ml gentamicin (all from Invitrogen Life Technologies). A375 and COLO205 were obtained from ATCC. A375 and COLO205 were, respectively, maintained in DMEM and in RPMI1640 with 10% FBS. All media were supplemented with 1% penicillin/streptomycin and 2 mM glutamine. The cells were grown according to standard protocols in a 37°C humidified, 5% $CO_2$ incubator. All cell lines were regularly verified as mycoplasma-free using a PCR-based test (Minerva Biolabs). Cell lines were authenticated by short tandem repeat profiling (Cell ID System; Promega) at 10 different loci (D5S818, D13S317, D7S820, D16S539, D21S11, vWA, TH01, TPOX, CSF1PO, and amelogenin).

PLX4720 was purchased from Selleck Chemicals, dissolved in DMSO at a final concentration of 500 mM and stored in aliquots at −80°C. Clinical-grade bevacizumab (Avastin, Roche) was stored as a stock solution of 25 mg/ml.

### Immunofluorescence analyses

Tumors samples were frozen in OCT compound and cut into 10-μm-thick sections after overnight treatment at 4°C in 30% sucrose solution. Tissue slices were fixed in 4% paraformaldehyde for 10 min at room temperature. The antibodies used for specific human or murine tissue immunostaining were as follows: anti-Ki67 (#MA5-14520; Invitrogen); anti-CD31 (#550274; BD Pharmingen); anti-NG2 (#AB5320; Millipore); anti-CD45 Alexa Fluor 647-conjugated (#103124; Biolegend); anti-CD68 (#ab125212; Abcam); anti-F4/80 (#MCA497A488; Bio-Rad); anti-collagen I (#ab21286; Abcam); anti-LOX (#ab31238; Abcam); anti-α-SMA Cy3-conjugated (#C6198; Sigma); anti-Gr1 (#108402; BioLegend); anti-Nrp1 (#AF566; R&D Systems); anti-CCR7 (#NB100-712; Novusbio); anti-CD206 Alexa Fluor 488-conjugated (#141710; Biolegend); and anti-CD11b PE-conjugated (#12-0112-82; Affymetrix eBioscience). The sections were then incubated with the appropriate fluorescence-conjugated secondary antibodies (Alexa 647 or 488, Life Technologies), and nuclei were counterstained with DAPI (Life Technologies). The samples were mounted using fluorescent mounting medium (Dako). All immunofluorescence images were captured and analyzed using a Leica SPEII confocal laser-scanning microscope (Leica Microsystems). Image acquisition was performed maintaining the same laser power, gain, and offset settings. Multiple independent fields (15–20 for every sections; 20× or 40× magnification) per tumor section were randomly chosen and analyzed from at least three tumors for each experimental condition. Image quantification was performed using NIH ImageJ and expressed as the fluorescence area. The MVD and MVA were quantified by counting CD31$^+$ vessels per mm$^2$ and measuring the CD31$^+$ area in the total field area. CD31$^+$ vessels were hand traced using ImageJ image analysis software to calculate the lumen vessel perimeter and area. Perivascular pericyte coverage

was quantified by double staining of CD31 and NG2. Cell apoptosis was assessed by terminal deoxynucleotidyl transferase dUTP nick-end labeling (TUNEL) assay (Roche) according to the manufacturer's specifications and by M30 CytoDEATH antibody (Roche), which recognizes a cytokeratin 18-specific cleavage product of the caspase cascade. All figures are representative of a tumor with a size included the in median size of the group analyzed.

## Tumor hypoxia and perfusion

Tumor hypoxia was detected by the formation of pimonidazole adducts at 1 h after intraperitoneal (i.p.) injection of 60 mg/kg pimonidazole hydrochloride in tumor-bearing mice. Mice were then sacrificed, and the tumors were harvested. Frozen sections were immunostained with hypoxyprobe-1-FITC-conjugated antibody (#MAb1; Hypoxyprobe Inc), according to the manufacturer's instructions. To visualize the functional blood vasculature, 200 μl of orange fluorescent microspheres (FluoSpheres; Life Technologies) was administered intravenously 10 min before tumor excision. The tumors were then frozen in OCT for immunofluorescence analysis and stained for CD31. The microsphere distribution was visualized by confocal microscopy using 10-μm sections, and tumor perfusion was analyzed by counting the % of double-positive blood vessels. Ten sections were analyzed per tumor.

## In vivo tumor growth and lung colonization assays

A375 ($10^7$) or COLO205 ($5 \times 10^6$) cells were subcutaneously injected in 6-week-old CD1 female athymic mice (Charles River). Tumor size was measured with a caliper, and tumor volume was calculated by the modified ellipsoid formula: length × (width)$^2$/2. When tumors reached a volume of approximately 250–300 mm$^3$, mice were randomly assigned to different treatment groups, which were maintained for 2 or 6 weeks. During randomization, mice with a tumor size lower or higher than 250–300 mm$^3$ were excluded.

For lung colonization assay, $5 \times 10^5$ MC-1 cells were injected into the tail vein of 6-week-old CD1 female mice and the animals mice were randomly assigned to different groups 12 weeks later. Micrometastases were blindly evaluated by light microscopy on paraffin-embedded and HE-stained slides. Five sections per tumor were analyzed.

Mice were treated with PLX4720 (daily oral gavage at 60 mg/kg, dissolved in a vehicle of 1% w/v methylcellulose in sterile water), bevacizumab (3 times a week by i.p. administration at 10 mg/kg, diluted with sterile 0.9% saline), the combination of PLX4720 and bevacizumab (COMBO), or equivalent volumes of vehicles.

To identify all significant therapeutic responses with a statistical power of 95%, we calculated the mice sample size by considering: (i) volume variations of ± 40% as random; (ii) volume variation of at least 50% as a therapeutic effect; (iii) a statistical significance of $P = 0.05$. Accordingly, at least five mice/group were used. Animal procedures were approved by the ethical committee of the University of Turin and by the Italian Ministry of Health.

## Microarray experiments

For gene expression profiling analysis, 300 ng of total RNA was amplified and labeled using an Illumina TotalPrep RNA Amplification Kit (Life Technologies). A total of 750 ng of 1,500 ng of labeled cRNA probes was hybridized on the HumanHT-12 v4.0 or MouseWG-6 v2.0 Expression Bead Chip, respectively (Illumina). All experiments were performed in biological triplicate (duplicate for COMBO group). Cubic spline-normalized probe intensity data, together with detection $P$-values, were obtained using the Genome-Studio software V2011.01 (Illumina). We selected probes characterized by at least one experimental point having a detection $P$-value < 0.05. For each gene, we retained the associated probe with the largest mean expression value across all samples. The data were log$_2$ transformed for all subsequent analyses. For each probe, the log$_2$ signal in each sample was converted to the log$_2$ ratio against the global average expression of that probe in all samples. The log$_2$ ratio expression data were clustered and visualized using GEDAS software (Fu & Medico, 2007). One-way ANOVA test and LIMMA (Smyth, 2005) were used to identify the modulated genes. For the ANOVA test, $P$-values were corrected for multiple testing using a Benjamini–Hochberg procedure (Benjamini & Hochberg, 1995), and a cutoff of 0.1 for the false-discovery rate (FDR) was used to define differentially modulated genes across all four experimental conditions. In the LIMMA analysis, a threshold of | log$_2$ FC | of 1 and $P$-value < 0.01 were used to select genes modulated to the greatest extent; each of the treatments was compared separately to the control samples. The data were analyzed for enrichment in biological themes (GO—Biological Processes) using the Database for Annotation, Visualization, and Integrated Discovery Bioinformatics Resources (DAVID, https://david.ncifcrf.gov).

Gene set enrichment analysis was performed according to the public application from the Broad Institute (http://www.broadinstitute.org/gsea/msigdb/). In particular, after gene filtering, for all the datasets, probes were collapsed on Gene Symbols, again selecting for each Gene the probe with the largest mean expression across all the experiments. GSEA Enrichment statistics were performed with the default setting, based on Pearson metric. $P$-values and FDRs were calculated by repeating sample permutations 1,000 times.

Ingenuity pathways analysis (Ingenuity Systems) was used to generate gene networks of significantly regulated groups of genes and identify related significant biological functions. Melanoma A375 xenograft microarray data have been deposited in The Gene Expression Omnibus of the National Center for Biotechnology Information (accession number GSE69754).

## Quantitative real-time PCR

Total RNA was extracted from tumors using an RNeasy mini kit (Qiagen). For cDNA synthesis, a High-Capacity cDNA Reverse Transcription kit (Life Technologies) was used according to the manufacturer's instructions. An RNA quality check, including concentration and purity, was performed with a Nanodrop ND-100 (Nanodrop Technologies). Quantitative real-time PCR was performed on a CFX96 (Bio-Rad) using SYBR-green PCR MasterMix (Life Technologies). The PCR thermal profiles were 95°C for 15 s and 60°C for 60 s (40 cycles). All experiments were performed in duplicate. Melting curve analysis was performed for each PCR to confirm the specificity of the amplifications. The housekeeping gene TBP was used to normalize the expression data. The primer sequences and product sizes are indicated in Appendix Table S6.

## Western blotting

Whole tumor samples were lysed in 1 ml of EB buffer (150 mM NaCl, 5 mM EDTA pH 8, 10 mM Tris–HCl pH 7.5, 1 mM $Na_3VO_4$, 1 mM Phenylmethylsulfonyl fluoride, protease inhibitors cocktail, 10% Glycerol, 1% Triton X-100) and dissociated by gentleMACS dissociator (Miltenyi Biotec). Samples were clarified by centrifugation for 10 min at 10,000 $g$, and proteins (30 μg) were resolved in a 4–20% SDS–PAGE gels and transferred onto PVDF membranes. Blots were incubated overnight at 4°C with mouse anti-TGFβ (#ab66043; Abcam) or rabbit antivinculin (#V9131; Sigma) antibodies (both at 1:1,000). After washing, membranes were incubated with horseradish peroxidase-conjugated secondary antibodies (1:10,000; Jackson). The immunoreactive bands were visualized by enhanced chemiluminescence detection kit (Perkin-Elmer Life Science Products) with the Chemidoc™ Touch Imaging system (Bio-Rad Laboratories, Inc). Band quantification was performed using Bio-Rad Quantity One software.

## Flow cytometry

Tumor lysates were prepared by mincing the tumor in DMEM (Sigma) and incubating in collagenase (Roche) at 37°C for 1 h. After washing in PBS, the cells were filtered through 70-μm filters (BD Biosciences). 5 × 10 cells were re-suspended in HBSS (Hank's balanced salt solution, Lonza) supplemented with 0.5% BSA (Sigma). Staining was performed at 4°C for 20 min, with the following antibodies: anti-mouse CD45 (clone 30-F11, Biolegend San Diego, CA, USA), anti-mouse F4/80 (clone BM8, Biolegend San Diego, CA, USA), anti-mouse cd11c (clone N418, Biolegend San Diego, CA), anti-mouse I-A/I-E (clone M5/114.15.2, Biolegend San Diego, CA, USA); anti-mouse TNFα (clone MP6-XT22, BD Pharmingen), anti-mouse Ly6C (clone HK1.4, eBioscience; San Diego, CA, USA); anti-mouse NOS2 (clone 6/iNOS/NOS Type II, BD biosciences, San Diego, CA, USA) and mouse IgG2a K (BD biosciences, San Diego, CA, USA). For intracellular staining, Cytofix/Cytoperm and Permwash staining kit (BD Pharmingen) were used. Cells were detected using the BD FACS Canto II cytofluorimeter and analyzed with FlowJo software.

## Cell co-culture

To evaluate the *in vitro* cytotoxic effect of the tumor cell population on A375 cells, the cells were infected with pLVX-ZsGreen1-N1 lentivirus (Clontech). Vector production was achieved in HEK-293T cells transfected with pLVX-ZsGreen1-N1 and packaging plasmids. Selection was achieved by a 24-h cell incubation with puromycin (2 μg/ml). Tumor cells (5 × 10^4) recovered as described in the "Flow cytometry" section were co-cultured with Zs-green-A375 cells (5 × 10^4) in 96-well plates (final volume, 70 μl). Fluorescence was recorded at day 0 and days 2 by fluorescence microscopy (Leica DMI3000B) and analyzed by NIH ImageJ software.

## Macrophage depletion

Macrophage depletion was obtained by treating mice 1 week after A375 cell implantation with i.p. injection of clodronate liposome (200 μl) (Encapsula Nanosciences, Brentwood, TN, USA) and

## The paper explained

### Problem
In the last 20 years, cancer treatment is significantly improved by the availability of drugs specifically targeting mutant oncogenes. However, the success of target therapies is usually dampened by the appearance of acquired resistance and disease relapse. More than half of melanomas are characterized by an activating BRAF mutation and transiently respond to BRAF[V600E] inhibitors, but almost all patients develop resistance to these agents and subsequently relapse. Therefore, there is a cogent clinical need to design new strategies to control and delay the onset of resistance.

### Results
The results of clinical trials using anti-angiogenic regimens to restrain tumor progression have been somewhat disappointing. However, judicious inhibition of VEGF promotes vascular normalization and favors the delivery of small molecules to tumors. Therefore, a promising application of anti-angiogenic drugs is their combination with other therapeutic strategies. We tested a combination of bevacizumab and the vemurafenib analog PLX4720, which target BRAF[V600E], and observed *de novo* transcriptional profile that sustains decrease of tumor hypoxia, accumulation of antitumor macrophages, and reduction in cancer-associated fibroblasts. These unexpected effects delayed the onset of resistance to PLX4720.

### Impact
Our findings offer a new perspective for the management of clinical resistance to BRAF inhibitors. Furthermore, the observation that the association of BRAF inhibitors with bevacizumab elicits the recruitment of antitumor macrophages supports the concept of combining this strategy with immune-based therapies.

repeated three times per week up to the end of COMBO treatment.

## Statistical analyses

The data are presented as the mean ± SEM. Student's *t*-test (two tailed) or one-way ANOVA, followed by *post hoc* pairwise analysis test, was performed using GraphPad Prism version 6.00 (GraphPad Software, La Jolla, CA, USA, www.graphpad.com). *P*-value < 0.05 was considered significant. The stars in the graphs indicate significance, as detailed in the figure legends.

**Expanded View** for this article is available online.

## Acknowledgements
This work was supported by Associazione Italiana per la Ricerca sul Cancro (AIRC) investigator grants IG (14284, 10104, 15585, 18652) and AIRC 5 × 1,000 (12182); Fondo Investimenti per la Ricerca di Base (codes: RBAP11BYNP, RBFR08F2FS-002), Fondazione Cassa di Risparmio di Torino and University of Torino-Compagnia di San Paolo. The authors thank Stefania Giove e Fabrizio Maina for technical support.

## Author contributions
VC, DC, FDN, DS, and FB contributed to conception and design; VC, EMi, EMe, FO, and FMC helped in development of methodology; VC, DC, FO, and FMC contributed to acquisition of data; VC, DC, FDN, DS, AS, DT, and FO helped in analysis and interpretation of data; VC, DC, FDN, DS, DT, AS, FO, and AB

contributed to writing, review, and/or revision of the manuscript; and FB supervised the study.

## Conflict of interest

The authors declare that they have no conflict of interest.

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
