## [Review Process File · EMBO Molecular Medicine]

VEGF BLOCKADE ENHANCES THE ANTITUMOR EFFECT OF BRAF^{V600E} INHIBITION

Valentina Comunanza, Davide Corà, Francesca Orso, Francesca Maria Consonni, Emanuele Middonti, Federica Di Nicolantonio, Anton Buzdin, Antonio Sica, Enzo Medico, Dario Sangiolo, Daniela Tavernam, and Federico Bussolino

Corresponding author: Federico Bussolino, University of Torino

Review timeline:

Submission date:	26 August 2015
Editorial Decision:	16 October 2015
Revision received:	16 October 2015
Editorial Decision:	19 October 2015
Revision received:	21 September 2016
Editorial Decision:	26 October 2016
Revision received:	08 November 2016
Accepted:	15 November 2016

Transaction Report:

Editor: Roberto Buccione

1st Editorial Decision

16 October 2015

Thank you for the submission of your manuscript to EMBO Molecular Medicine. We have now heard back from the three Reviewers whom we asked to evaluate your manuscript.

We are sorry that it has taken longer than usual to get back to you on your manuscript. We experienced some difficulties in securing the evaluations in a timely fashion.

As you will see, the very expert Reviewers point to significant, fundamental and mostly overlapping issues that, I am afraid, preclude publication of the manuscript in EMBO Molecular Medicine. I will not discuss each point in detail as they are clearly stated.

In brief, and in aggregate, the issues include on one hand the lack of essential novelty also with respect to your own previous work (and where elements of novelty are present, there is insufficient mechanistic analysis). The other major concerns are limitations of the experimental model applied with unconvincing pre-clinical experimentation and consequently unconvincing case for translational value. Some examples are for instance, unclear effect of the combination therapy and the individual constituents on EC and immune cells, the short therapeutic window, and I also agree with the limitations of the study in terms of pre-clinical value.

After much internal discussion and Reviewer cross commenting, it was agreed that the findings are on the whole interesting but that the time required to fully satisfy the legitimate concerns would be

much more than we would find acceptable for a revision (also due to the animal studies required) also because of their inherent complexity and challenging nature.

Given these fundamental concerns, I have no choice but to return the manuscript to you at this stage. In our assessment it is not realistic to expect to be able to address these issues experimentally and to the satisfaction of the Reviewers in a reasonable time frame.

I wish to add however, that considered the potential interest of these findings, we would have no objection to consider a new manuscript on the same topic if at some time in the near future you have obtained data that would considerably strengthen the message of the study and address the Reviewers' concerns. If you decide to follow this route, please make sure you nevertheless upload a letter of response to the referees' comments.

I am sorry to have to disappoint you at this stage. I hope that the Reviewers' comments will be helpful in your continued work in this area.

***** Reviewer's comments *****

Referee #1 (Comments on Novelty/Model System):

There is excessive reliance on immunohistochemistry. Essentially, the paper employs three methods, immunostaining, expression profiling and PCR. There is not functional or mechanistic analysis. Since there appears to be strong contribution of the immune component, the use of immune deficient mice is not fully adequate.

Referee #1 (Remarks):

The study by Comunanza et al dissects the combined effect of BRAF-targeting therapy and anti-angiogenic VEGF blocker bevacizumab on the melanoma growth in mice. Using subcutaneous xenografts as a model, they demonstrate that in a 2-week window each agent individually stabilized tumor growth, while the combination caused partial tumor regression. To investigate the mechanisms underlying this cooperative activity the authors undertake a combination of extensive IHC analyses and expression profiling.

- The IHC findings are quite similar between PLX4720, and the PLX4720/bevacizumab combination. There are similar decreases in proliferation (Fig. 1), CD31-positive microvascular area and the size of the vascular lumen (Fig. 2). The only discernible difference is the increased rate of keratin-14 cleavage, presumably by Caspase-3 (Fig. 1). It is unclear why more conventional assays for apoptosis have not been used to confirm this finding. The perfusion rate and hypoxic index is also roughly similar for these two treatment conditions and demonstrate the increased perfusion and decreased hypoxia (Fig. 3). The results regarding tumor necrosis are impossible to interpret as shown: better quality images are needed to discern necrosis from fibrosis, but in any event there is no significant difference between PLX4720, and the PLX4720/bevacizumab combo at a glance (Fig. 3).
- Expression profiling also shows limited differences between the tumors treated with PLX4720, or the combo, suggesting that the effect of bevacizumab on the tumor cells is minor (a predictable outcome). In both cases, the genes linked with immune and inflammatory responses, ECM remodeling and adhesion are increased in a similar pattern (Figs. 4 and 5).
- The authors focus on immune response and matrix remodeling and show a major increase in the presence of CD45+ and F4/80 positive cells suggesting the increased infiltration of monocytes and macrophages caused specifically by the combination treatment. Further characterization points to some M1-like features of the infiltrating macrophages; however, all the results were obtained by either immunostaining or by Q-PCR analysis of the entire tumor and not of the isolated macrophage population. In addition, there is an increase in neuropilin-1 positive monocytes, which are thought to possess anti-tumor properties.
- These are all very interesting and compelling findings, however the paper has one but pervasive

flaw one flaw, the complete lack of the mechanistic explanation for this increased antitumor immune response caused by the combined PLX4720/bevacizumab treatment, especially given only minor differences in the expression profile. In such a case one suspects a cell-type specific response, which should be tested on the cultured cells and using micro-dissection of the tumors followed by qPCR or FACS analysis, microfluidics and other modern methodologies. As it stands now, it appears that the dramatic effect of the combination therapy is channeled through a specific cell population, which is yet to be conclusively identified.

- Given the prevailing immune component of the response, the analysis of the circulating monocytes and macrophages as well as the bone marrow populations would greatly enhance the study.
- Although it is extremely likely, there is no formal proof that the observed increased macrophage infiltration is responsible for the remarkable differences in efficacy. This question could be easily answered using macrophage-depleted mice. In general, functional and mechanistic studies are absent from this study.
- Given significant contribution of the immune system the use of immune compromised mice may be not adequate.
- The potential role of the NRP-1 positive monocyte population could also be ascertained experimentally.
- Finally, it is unclear what are the molecular driver(s) of the major decrease in fibrotic response (seen in Fig. 7).

In conclusion, the mechanistic aspect of this very interesting study is underdeveloped and publication in EMBO MOLECULAR Medicine appears premature at this stage.

Referee #2 (Remarks):

Background: The authors present preclinical data to support the idea that combining an antiangiogenic drug that targets the VEGF pathway, e.g. bevacizumab - the VEGF antibody - with an agent that blocks the function of the BRAFV600E mutation such as vemurafenib (or in this case, an agent called PLX4720 - another oral drug similar to vemurafenib) for the treatment of malignant melanoma. The authors report evidence using a couple of primary melanoma subcutaneous transplant models that the efficacy of the combination treatment (called "COMBO") is more effective than either drug alone. They then go on to provide a series of detailed mechanistic studies focusing on such things as vessel normalization, perfusion, and altered gene signatures, which may explain the basis of the improved efficacy of this treatment combination. In this regard, the authors' mechanism studies are quite detailed and extensive (in contrast to their in vivo therapy models and studies).

Critique: This is an interesting and potentially important study, and in some ways, it is also ironic. I say this because the clinical development and assessment of such a drug combination in melanoma patients has already been initiated, and this essentially took place in the absence of compelling detailed preclinical studies and rationale. I'll have more to say about this below, but thus far the clinical studies have been disappointing, primarily because of toxicity.

While the notion of combining an anti-VEGF agent with another drug that targets the BRAFV600E mutation would seem logical and timely to evaluate, I have a major reservation and criticism of the work presented by the authors. While I think the detailed mechanism studies, which in essence constitute the bulk and thrust of the paper, are interesting and reasonably sound, they were obviously undertaken because of the supposed encouraging efficacy/therapy results reported in Fig. 1 and Suppl Fig. 5. This is where I have a major problem. In Fig. 1 the authors undertake an extremely short term therapy - less than 2 weeks! - on established primary tumors and show a rather modest degree of anti-tumor efficacy. In Suppl Fig. 5 they use a longer period of therapy, the duration of which, however, is not entirely clear to me. The experiment lasted 6 weeks but it is not clear exactly when administration of the drugs was initiated. In addition, in both such experiments the number of mice per group was around 5 or 6, and in Suppl Fig. 5, at the conclusion of the experiment the curves were beginning to look like they were shortly going to converge, showing ultimately no benefit. This might be similar to seeing a progression free survival benefit in the clinic, but not an

overall survival benefit. Indeed, this is another issue - tumor volumes and growth delay effects were measured, but not survival.

This kind of *in vivo* therapy analysis, in this reviewer's opinion, is no longer acceptable, or at least meets a high scientific standard of the kind that would be expected of publication in a journal such as EMBO Mol Med. The concern is that what is being seen in the subcutaneous primary tumor transplants will not necessarily be reflected in more clinically relevant situations such as treatment of distant metastases. In melanoma patients this can occur almost anywhere, but particularly in the liver, lungs, and brain. Would such metastases, especially overt established metastases, respond to the COMBO in a significant and better way than either drug alone? Melanoma patients virtually never receive neoadjuvant therapy for treatment of their primary tumors. These tumors are exceedingly small and they are always removed by wide excision surgical means. Therefore, the absence of any data evaluating the combination in a metastatic setting is a major deficiency of this study. In addition, nowadays it is virtually mandatory for a high quality paper published in an extremely good journal to have more up-to-date and clinically relevant models. Here I'm referring to the use of patient-derived xenografts (although this would involve treatment of primary tumors only) and genetically engineered mouse models of cancer which, in all likelihood, would also involve treatment of established primary tumors. Nevertheless, I think some PDX analyses would add weight to the overall result if the combination treatment showed meaningful efficacy that were superior to either drug alone.

I'm sure the authors are well aware of the fact that time and again throughout the course of the history of experimental therapeutics in oncology, drugs or therapies found effective in mice failed in the clinic. One of major reasons why is toxicity. This brings me to the short duration of the therapies undertaken by the authors, and the drugs used. Thus, the authors use bevacizumab which only neutralizes human VEGF and which can only come from the human tumor transplants, but there is a lot of VEGF in mice which of course is derived from mouse cells/tissues; neutralization of this endogenous VEGF can lead to side effects in humans, which can sometimes be serious. The class effects are well known such as hypertension, proteinurea, clotting and bleeding, etc. Moreover, these side effects may be exacerbated by combination with another drug. The short term nature of the therapy protocols along with the use of an antibody that neutralizes only human VEGF will no doubt mask or even totally prevent any potential toxicity from being observed. Thus longer term experiments, especially those evaluating metastatic disease, would seem to be in order here, and doing so with an agent that is not restricted to neutralizing only human VEGF.

I mentioned earlier that there has already been an attempt previously to evaluate a drug such as vemurafenib plus or minus bevacizumab in patients with advanced metastatic melanoma. Indeed, a trial in the US was launched about two years ago based mainly on some limited preclinical and clinical data that appeared to support the notion that VEGF may be contributing to the development of resistance to BRAF inhibitor therapy. This trial had to be stopped and the protocol revised because of the superiority of the combination of a BRAF inhibitor with an MEK inhibitor (cobi) that had been observed in various other clinical trials. More recently, a trial of vemurafenib plus cobimetinib plus or minus bevacizumab was initiated, but this trial is experiencing difficulty too because of toxicity and it is not clear how far it will go. In addition, across the major melanoma centres in the US there is a phase II randomized trial being conducted of a combination of a BRAF inhibitor, a MEK inhibitor, plus or minus bevacizumab. Initiation of this trial was apparently supported by borderline positive randomized phase II clinical trial data of chemotherapy (carboplatin plus paclitaxel plus or minus bevacizumab).

All in all, there has never been a compelling body of preclinical evidence to suggest that the combination of a BRAF inhibitor plus an antiangiogenic drug such as bevacizumab would be an effective therapy in metastatic melanoma patients. However, neither is there evidence to suggest that this would not be an effective therapy, and here is where the authors have a potentially interesting study. However, as mentioned throughout this review, I think the preclinical results do not yet justify the conclusion that this would be an effective therapy for the treatment of metastatic melanoma in the clinic. Therefore my recommendation would not be to reject this manuscript, but neither, obviously, can it be considered for acceptance. It needs major revisions and these revisions should involve more rigorous evaluation of the combination therapy involving better and more up-to-date models, as suggested above, and involving longer duration therapy with appropriate drugs that target not only human VEGF, etc. Should these models continue to show evidence in support of

what the authors suggest - that these two types of drug may be effective in treating BRAF mutant melanoma - this could turn out to be a seminal contribution to the cancer therapeutic literature. I realize that such experiments could take some time, but in view of my concerns I see no alternative.

One last point, and that is that I think the authors have distorted the supposed value of vessel normalization as a mechanism to increase therapy efficacy. There are increasing numbers of studies coming out, both preclinical and clinical, that an antiangiogenic drug actually diminishes the intratumoral delivery of concurrently administered drugs including chemotherapy or other antibodies such as trastuzumab. Thus, I think the authors should be more careful in their assessment of the supposed benefits of vessel normalization as a clinically relevant mechanism for increasing cancer therapy efficacy. This concern is also mentioned because, once again, all of the mechanism results were undertaken on transplanted subcutaneously grown primary tumors, and I am extremely dubious whether the results would pertain in a general way to melanomas growing in metastatic sites such as the lungs, liver, or brain.

Recommendation: Reconsider after major revisions.

Referee #3 (Comments on Novelty/Model System):

I am a confused about whether this is specific to BRAF mutant tumors and think this can be directly addressed by the authors and I put this in my comments. This hopefully is an easy animal expt to perform and regardless of the results it will inform the reader/author about the mechanism of the effect observed.

Referee #3 (Remarks):

Review for: VEGF blockade enhances the antitumor effect....

Comunanza et al initial submission.

This manuscript provides evidence that combination of bevacizumab and the BRAF inhibitor PLX4720 is an effective anti-cancer strategy. In fact the study shows that the combination has surprising therapeutic efficacy in preclinical models of melanoma and colon cancer. The study builds on prior work from the group on Braf inhibition and its effect on tumor vasculature. Further, there are a number of intriguing aspects to the study that elevate the work beyond the typical combinatorial study. The authors detail surprising changes in hypoxia, ECM remodeling and fibroblast activation after combination of bevacizumab with PLX4720. A strong point of the study is the transcriptional analysis the authors perform on tumors from mice treated with single agent and combination therapy. Overall, given the fact that the data seem to run counter to what would be expected with an anti-angiogenic agent in combination with a kinase inhibitor, the burden of evidence is high for the conclusions drawn. Thus while the results are intriguing there are a number of challenges that should be addressed. As presented the conclusions appear to be the result of over interpretation of the data presented.

Major concerns:

1. It is not surprising that blocking VEGF and BRAF activity is more effective in controlling tumor growth but the changes in tumor perfusion and hypoxia after combination therapy are surprising. I realize this was reported in the PNAS 2012 paper but in that paper single agent therapy was used and the mechanism of how PLX4720 alters perfusion and hypoxia is not clear.
 - a. The PNAS 2012 paper states that PLX4720 does not directly affect endothelial cells but reduces proangiogenic molecule expression by tumor cells. However, the authors should look at endothelial cell proliferation in the tumors treated with PLX4720 and combo; this could be done by IHC for a proliferation marker and an endothelial cell marker. This would help clarify how PLX4720 increases perfusion and reduces hypoxia.
 - b. Is this surprising effect of combination therapy limited to BRAF mutant tumors? There are numerous studies in multiple tumor models showing that anti-angiogenic therapy in combination with another agent increases hypoxia and reduces perfusion, thus the data presented here are quite distinct and suggest that PLX4720 has activity beyond tumor cells. If the effect is solely due to a reduction in secretion of factors from tumor cells, which factors are most critical to the stromal changes? Can the authors speculate and provide some protein data to validate that speculation. They do not need to functionally validate the candidate soluble factors.
 - c. Acute therapy with an anti-VEGF can result in transient vascular normalization. However there

are a few caveats. First bevacizumab only inhibits human VEGF, thus in the xenograft setting mouse VEGF secreted from the stromal cells in the tumor microenvironment is still functional. The authors should evaluate the level of mouse VEGF protein, is it possible that PLX4720 reduces mouse VEGF production/secretion?

d. Chronic therapy should be explored. The data presented in Figure 7 speak to this somewhat. What happens to vascular density in the responding and relapsing tumors?

2. The other surprising changes observed in tumors treated with combination therapy are the dramatic changes in immune cell recruitment and fibroblast activation and the change in collagen deposition.

a. Again the question of the effect of PX4720 on stromal cells is relevant. Does PLX4720 directly affect immune cells or fibroblasts or are the changes in the tumor microenvironment due to the alleviation of hypoxia? This should be addressed with at minimum in vitro studies evaluating the effect of PLX4720 on a fibroblast cell line and the discussion should be expanded in this area.

b. The authors should directly evaluate TGF β protein as this growth factor is often prevalent in solid tumors and relevant to the phenotypes they are reporting (e.g., fibroblast, immune cells, and collagen deposition).

3. A therapy study with the same treatment groups in a non-BRAF mutant tumor should be provided.

4. Can the authors speculate on potential biomarkers that could be evaluated in patients treated with a BRAF inhibitor that would predict response or justification for combination with another therapy?

5. In general the number of animals in each experiment should be clarified. Further it would be useful to the reader to report the tumor volume of the tumors being compared by IHC or functional analysis (e.g., perfusion/hypoxia). This is especially relevant for the responding v relapsing tumors. Minor concerns

1. Why is CD146 used for evaluation of the vascular compartment in Colo205 xenografts. The images with CD146 look quite distinct from the CD31 used for MVD studies in the melanoma sections. It is hard to compare the effect between the two xenografts when different vascular markers are used.

2. The images used for PLX4720 and Bev in panel 6E are the same

3. F4/80 is misspelled - line 5 page 12

1st Revision - authors' response

16 October 2015

Thanks for the opportunity to re-submit as a "de novo" manuscript our paper 05774 entitled "VEGF blockade enhances the antitumor effect of BRAFV600E inhibition".

The referees pointed many experiments that will improve the quality of the MS and in principle they will allow clarifying some points not well addressed in the previous version.

The purpose of this letter is to explain and discuss what we think to be addressable in a reasonable time frame to reply to the major referees' criticisms.

We'll perform the most of the experiments requested. However I believe that some of them are not strictly necessary and do not improve the meaning of the paper. However, I'd appreciate your opinion about these specific proposals, in order to help us to perform the best revision.

Referee #1

1) A first point underlined by this referee in his/her general remarks is the fact that "the use of immune deficient mice is not fully adequate". I agree with this criticism but the currently available tools seem to be inappropriate. Actually a BRAFV600E transgenic mice is accessible. It develops hyperplasia of melanocytes and Schwann cells but less than 10% of the mice show melanomas. The disease onset is largely variable and related to the abundance of the mutated transcript (Goel VK, et al Melanocytic nevus-like hyperplasia and melanoma in transgenic BRAFV600E mice. Oncogene.

2009 28:2289). Furthermore, this model is not appropriate for the use of bevacizumab, which recognizes human VEGF.

Therefore I'll stress in the text the limit of our approach that does not take into account the role of lymphocytes but it considers only the myeloid compartment.

- This referee asks to analyze the apoptosis with another technique.

We'll do that.

-He/she criticizes the necrosis analysis. *We'll delete this picture because it is not so important.*
Actually the effect on necrosis depends on the effect on perfusion and hypoxia alleviation induced by PLX4720 and COMBO, that have been analyzed

-He/she was impressed by the relevant effect exerted by COMBO on leucocyte infiltrate and suggested to conclusively identify the effector population, which we suggest to be M1-macrophages. The following experiments are planned: 1*) effect of COMBO in monocyte-depleted mice by clonodrate; 2) immune-phenotype of CD45 cell isolated from tumors by FACS analysis.*

In our current version we demonstrated the cytotoxic effect of bulk cell population of COMBO-treated tumors on GFP-A375 cell line. We 'll plan to repeat the experiments with purified myeloid cells from the tumors.

- The analysis of the circulating monocytes and macrophages as well as the bone marrow populations would greatly enhance the study.

We'll do that (at least for circulating cells)

-The potential role of the NRP-1 positive monocyte population could also be ascertained experimentally. We described that COMBO increases this myeloid subpopulation. However it's a very minute population and the increased promoted by COMBO is not so high. In the case of clear data obtained by analyzing the effect of M1-population, *I suspect that the suggested experiments do not necessarily add values to the manuscript.*

It is unclear what are the molecular driver(s) of the major decrease in fibrotic response. *We'll analyze the expression (protein and mRNA) of pro-fibrotic molecules*

Referee #2

The major concerns of this referee are: i) the models are far from the clinical practice and he/she suggest to add a model that allows studying the effect of COMBO on metastasis, and to treat patient-derived xenografts by COMBO; ii) the models do not consider murine VEGF; iii) the adverse effects are not studied.

We'll plan to add a model of metastatisation and to *measure murine VEGF* and discuss the data obtained in the light of bevacizumab effect.

I don't think that PDX could represent a real advantage because the high rate of engraftment is usually obtained in NOD-SCID mice, in which myeloid compartment is almost absent. As far as concern the toxic effect, I think that this request is appropriate but it is far from the aim of the paper.

We can say that the treatments are well tolerated by the animals without any loss of weight or premature deaths.

Referee #3 Major concerns:

He/she asks to measure endothelial proliferation in the tumor differently treated.

We'll do that.

About the mechanisms: Can the authors speculate and provide some protein data to validate that speculation?

We'll do that

The authors should evaluate the level of mouse VEGF protein; is it possible that PLX4720 reduces mouse VEGF production/secretion?

*We'll do that *

What happens to vascular density in the responding and relapsing tumors?

We'll do that

Again the question of the effect of PX4720 on stromal cells is relevant. Does PLX4720 directly affect immune cells or fibroblasts or are the changes in the tumor microenvironment due to the alleviation of hypoxia? This should be addressed with at minimum in vitro studies evaluating the effect of PLX4720 on a fibroblast cell line and the discussion should be expanded in this area.

We'll plan experiments to verify if a co-culture system (A375+cancer associated fibroblast +/- endothelial cells +/- monocytes) can recapitulate the reduction of CAF exerted by COMBO

A therapy study with the same treatment groups in a non-BRAF mutant tumor should be provided.

I agree that PLX4720 can have off-target effect (i.e. inhibition of c-RAF). However I think that to a complete and clinically relevant response needs the analysis of more models*.

I think this point could be only discussed.*

Other points we'll be easily addressed

I believe that these proposed experiments can improve our MS and satisfy the major requests of the referees

2nd Editorial Decision

19 October 2015

Thanks again for sending me a rebuttal and plan for further experimentation.

I apologise that it has taken a few days to respond but I needed to go back to your manuscript and the related correspondence. I have now managed to go over your letter and I believe that, although I cannot decide for the Reviewers, the points you make appear reasonable, including those you feel did not warrant further experimentation. Therefore, should your experiments be successful, your proposed revision should nicely address the concerns expressed by the reviewers. I urge you to carefully discuss each point upon re-submission.

I would just like to reiterate that we agree that to provide evidence in a "proper" metastasis model is crucial, regardless of whether this is pursued within a PDX setting, the limitations of which in your case I understand. In addition of course to the other clarifications that were requested.

I look forward to receiving a new manuscript in due time.

2nd Revision - authors' response

21 September 2016

I thank the referees for their useful suggestions and criticisms and point-to-point reply is here provided.

Referee #1 (Comments on Novelty/Model System):

There is excessive reliance on immunohistochemistry. Essentially, the paper employs three methods, immunostaining, expression profiling and PCR. There is not functional or mechanistic analysis.

We fully agree with these comments. The experiments of the initial manuscript have been integrated with the FACS analysis of myeloid cells present in tumor tissue, blot analysis of the expression of a TGF β and macrophage depletion by xenograft treatment with clodronate.

Since there appears to be strong contribution of the immune component, the use of immunodeficient mice is not fully adequate.

We are aware of this limit and at page 22 (lines 20-22) we added the following sentence:

'Further experiments will be required to investigate how T and B cells, which are absent in CD1 mice, modify the functional performance of the association between BRAFV600E targeting and VEGF removal.'

We believe that the absence of a syngeneic model does not reduce the relevance of this paper that shows the role of myeloid cells exclusively recruited by the combination of bevacizumab and PLX4720. Furthermore, to the best of our knowledge, there is a single murine cell line carrying BRAFV600E (doi: 10.1038/onc.2009.95), but it is unable to metastasize, and therefore it is not fully representative of effect of BRAFV600E in humans.

Referee #1 (Remarks):

The study by Comunanza et al dissects the combined effect of BRAF-targeting therapy and anti-angiogenic VEGF blocker bevacizumab on the melanoma growth in mice. Using subcutaneous xenografts as a model, they demonstrate that in a 2-week window each agent individually stabilized tumor growth, while the combination caused partial tumor regression. To investigate the mechanisms underlying this cooperative activity the authors undertake a combination of extensive IHC analyses and expression profiling. The IHC findings are quite similar between PLX4720, and the PLX4720/bevacizumab combination. There are similar decreases in proliferation (Fig. 1), CD31-positive microvascular area and the size of the vascular lumen (Fig. 2). The only discernible difference is the increased rate of keratin-14 cleavage, presumably by Caspase-3 (Fig. 1). It is unclear why more conventional assays for apoptosis have not been used to confirm this finding.

We followed the referee's advice and we analysed apoptosis by TUNEL staining (new Figure 1E) which confirmed the previous analysis done with Cyto Death kit (now Fig S1).

The perfusion rate and hypoxic index is also roughly similar for these two treatment conditions and demonstrate the increased perfusion and decreased hypoxia (Fig. 3). The results regarding tumor necrosis are impossible to interpret as shown: better quality images are needed to discern necrosis from fibrosis, but in any event there is no significant difference between PLX4720, and the PLX4720/bevacizumab combo at a glance (Fig. 3).

We agree with the reviewer that the H&E staining was poorly informative in showing any morphological differences between the treatment groups, and we propose to delete this panel in order to make space for the newly generated functional and mechanistic data.

Expression profiling also shows limited differences between the tumors treated with PLX4720, or the combo, suggesting that the effect of bevacizumab on the tumor cells is minor (a predictable outcome). In both cases, the genes linked with immune and inflammatory responses, ECM remodeling and adhesion are increased in a similar pattern (Figs. 4 and 5). The authors focus on immune response and matrix remodeling and show a major increase in the presence of CD45⁺ and F4/80 positive cells suggesting the increased infiltration of monocytes and macrophages caused specifically by the combination treatment. Further characterization points to some M1-like features of the infiltrating macrophages; however, all the results were obtained by either immunostaining or by Q-PCR analysis of the entire tumor and not of the isolated macrophage population.

We extended immunofluorescence and qPCR analyses by performing also flow cytometric assessment of cells isolated from A375 tumors. The new results are now reported in Figures 4 and 5 and confirm the previous results. Specifically, Figures 4C and 4D display that COMBO treatment induced a higher CD45⁺ leukocyte recruitment than vehicle, which was mainly constituted by F4/80⁺ TAMs and dendritic CD11c⁺/MHCII⁺ cells. FACS analyses were also performed to show that COMBO increased their number of tumor-recruited inflammatory monocytes, which are identified as CD45⁺ Cd11b⁺ Ly6C^{high} cells, in A375 tumors but not in bloodstream (Figures 5D and 5E) (page 13, lines 16-22). FACS analysis was also utilized to confirm the M1 polarization of TAM recruited in xenografts by COMBO (Fig 5C and 5D).

In addition, there is an increase in neuropilin-1 positive monocytes, which are thought to possess anti-tumor properties

See below

These are all very interesting and compelling findings, however the paper has one but pervasive flaw one flaw, the complete lack of the mechanistic explanation for this increased antitumor immune response caused by the combined PLX4720/bevacizumab treatment, especially given only minor differences in the expression profile. In such a case one suspects a cell-type specific response, which should be tested on the cultured cells and using micro dissection of the tumors followed by qPCR or FACS analysis, microfluidics and other modern methodologies. As it stands now, it appears that the dramatic effect of the combination therapy is channeled through a specific cell population, which is yet to be conclusively identified.

We thank the referee for his/her insightful comments. In order to address this appropriate criticism we applied two different strategies: i) macrophage depletion by clodronate treatment (Figure 7) and ii) the ex vivo stimulation of M1 switch induced by PMA and ionomycin (Figure 5). These experiments indicate that macrophage depletion markedly reduces COMBO effect and that TAMs present in COMBO-treated tumors are committed to an M1 phenotype. These new data connected with the previous results showing that M1-like TAMs continue to be present in responding but not relapsing mice (Fig S6B) suggest that this macrophage subset is instrumental in COMBO activity. We cannot exclude that NEM or other cells can contribute to COMBO effect. The following sentence is added in the Discussion (page 18, lines 21-24):

“A secondary effect of COMBO treatment on myeloid compartment is an increase of asubpolupation expressing neuropilin-1 and CD11b, which have been reported to favor vascular normalization and inhibit tumor growth when directly injected in B16.F10melanomas at relative high cell concentration (2×10^5 cells/tumor) (Carrer et al., 2012)”

However we believe that the little expansion of this compartment promoted by COMBO regimen is not necessarily pivotal in the observed tumor growth inhibition.

Given the prevailing immune component of the response, the analysis of the circulating monocytes and macrophages as well as the bone marrow populations would greatly enhance the study. •

We analysed the presence of inflammatory monocyte ($CD45^+ Cd11b^+ Ly6C^{high}$) in bloodstream and in tumors after COMBO treatment. The data reported in Figure 5 show that COMBO regimen has not a systemic effect but rather favors M1 polarization in tumor stroma (pag 13, lines 13-20).

Although it is extremely likely, there is no formal proof that that the observed increased macrophage infiltration is responsible for the remarkable differences in efficacy. This question could be easily answered using macrophage-depleted mice. In general, functional and mechanistic studies are absent from this study.

As mentioned above, in order to address this criticism, we have applied clodronate treatment to deplete macrophages. The results of these experiments are shown in Figure 7.

Given significant contribution of the immune system the use of immune compromised mice may be not adequate.

See our comments to the initial remarks of this referee

The potential role of the NRP-1 positive monocyte population could also be ascertained experimentally.

See our above comment

Finally, it is unclear what are the molecular driver(s) of the major decrease in fibrotic response (seen in Fig. 7)

In the old version IPA analysis allowed to predict a role of TGF β . This prediction has now been confirmed by assessing the level of this molecule in A375 tumors. New experiments now shown in Figure 6, show that COMBO does not modify the expression of murine Tgfb1 mRNA, but it is effective in strongly reducing the expression of the human genes (page 15, lines 15-22). In the discussion we added a comment on this issue (page 22, lines 5-15).

Referee #2 (Remarks):

While the notion of combining an anti-VEGF agent with another drug that targets the BRAFV600E mutation would seem logical and timely to evaluate, I have a major reservation and criticism of the work presented by the authors. While I think the detailed mechanism studies, which in essence constitute the bulk and thrust of the paper, are interesting and reasonably sound, they were obviously undertaken because of the supposed encouraging efficacy/therapy results reported in Fig. 1 and Suppl Fig. 5. This is where I have a major problem. In Fig. 1 the authors undertake an extremely short term therapy -less than 2 weeks!- on established primary tumors and show a rather modest degree of anti-tumor efficacy. In Suppl Fig. 5 they use a longer period of therapy, the duration of which, however, is not entirely clear to me. The experiment lasted 6 weeks but it is not clear exactly when administration of the drugs was initiated.

*We agree with this referee that preclinical experiments, including the drug schedule, are far from clinical settings and often are not useful to predict the human response. However, we used accepted schedules that take into account the species differences in term of life-span and pharmacokinetics and the experimental limits defined by ethical rules and experimental plan (i.e. subcutaneous injection of cancer cells). Finally, at page 25 (lines 17-22) we specified our experimental groups: **“When tumors reached a volume of approximately 250-300 mm³, the mice were randomly assigned to different treatment groups, which were prolonged for 2 or 6 weeks. Mice were treated with PLX4720 (daily oral gavage at 60 mg/kg, dissolved in a vehicle of 1% w/v methylcellulose in sterile water), bevacizumab (3 times a week by i.p. administration at 10 mg/kg, diluted with sterile 0.9% saline), the combination of PLX4720 and bevacizumab (COMBO), or equivalent volumes of vehicles”.***

In addition, in both such experiments the number of mice per group was around 5 or 6, and in Suppl Fig. 5, at the conclusion of the experiment the curves were beginning to look like they were shortly going to converge, showing ultimately no benefit. This might be similar to seeing a progression free survival benefit in the clinic, but not an overall survival benefit. Indeed, this is another issue -tumor volumes and growth delay effects were measured, but not survival.

The current ethical rules do not allow the assessment of the overall survival, because it is forbidden to wait the spontaneous mice death in order to avoid sufferance and pain.

This kind of in vivo therapy analysis, in this reviewer's opinion, is no longer acceptable, or at least meets a high scientific standard of the kind that would be expected of publication in a journal such as EMBO Mol Med. The concern is that what is being seen in the subcutaneous primary tumor transplants will not necessarily be reflected in more clinically relevant situations such as treatment of distant metastases. In melanoma patients this can occur almost any where, but particularly in the liver, lungs, and brain. Would such metastases, especially overt established metastases, respond to the COMBO in a significant and better way than either drug alone? Melanoma patients virtually never receive neo adjuvant therapy for treatment of their primary tumors. These tumors are exceedingly small and they are always removed by wide excision surgical means. Therefore, the absence of any data evaluating the combination in a metastatic setting is a major deficiency of this study.

To respond to the appropriate criticism on the lack of experiments considering the effect of melanoma metastatization, we added new experiments by using a metastatic sub clone of A375 cell line (MC-1 cell line) in a lung colonization assay. Figure 1 shows that COMBO regimen consistently reduces the lung area colonized by this cell line i.v. injected (page 5, lines 12-17).

In addition, nowadays it is virtually mandatory for a high quality paper published in an extremely good journal to have more up-to-date and clinically relevant models. Here I'm referring to the use of patient-derived xenografts (although this would involve treatment of primary tumors only) and genetically engineered mouse models of cancer which, in all likelihood, would also involve treatment of established primary tumors. Nevertheless, I think some PDX analyses would add weight to the overall result if the combination treatment showed meaningful efficacy that were superior to either drug alone.

In our opinion all kind of preclinical models are far from the clinical practice and have just to be considered partial or exhaustive proof of concepts of working hypotheses. This is also the case of this work that demonstrates a prominent role of myeloid cells, which are completely absent in NOD-SCID mice, the best recipient animals of primary human cancer cells. Actually the engraftment of primary human cells from different origins are poor in CD1 at hymic mice (doi: 10.1158/0008-5472.CAN-15-0727; doi: 10.1016/j.bbcan.2015.03.002).

I'm sure the authors are well aware of the fact that time and again throughout the course of the history of experimental therapeutics in oncology, drugs or therapies found effective in mice failed in the clinic. One of major reasons why is toxicity. This brings me to the short duration of the therapies undertaken by the authors, and the drugs used. Thus, the authors use bevacizumab which only neutralizes human VEGF and which can only come from the human tumor transplants, but there is a lot of VEGF in mice which of course is derived from mouse cells/tissues; neutralization of this endogenous VEGF can lead to side effects in humans, which can sometimes be serious. The class effects are well known such as hypertension, protein, urea, clotting and bleeding, etc. Moreover, these side effects may be exacerbated by combination with another drug. The short term nature of the therapy protocols along with the use of an antibody that neutralizes only human VEGF will no doubt mask or even totally prevent any potential toxicity from being observed. Thus longer term experiments, especially those evaluating metastatic disease, would seem to be in order here, and doing so with an agent that is not restricted to neutralizing only human VEGF.

We agree with this criticism and in our previous comment we claimed our awareness of the limits of preclinical models. However, the evaluation of COMBO toxicity is beyond the aim of this paper as well as the association of PLX2740 with other anti-angiogenic compounds.

I mentioned earlier that there has already been an attempt previously to evaluate a drug such as vemurafenib plus or minus bevacizumab in patients with advanced metastatic melanoma. Indeed, a trial in the US was launched about two years ago based mainly on some limited preclinical and clinical data that appeared to support the notion that VEGF may be contributing to the development of resistance to BRAF inhibitor therapy. This trial had to be stopped and the protocol revised because of the superiority of the combination of a BRAF inhibitor with an MEK inhibitor (cobi) that had been observed in various other clinical trials. More recently, a trial of vemurafenib plus cobimetinib plus or minus bevacizumab was initiated, but this trial is experiencing difficulty too because of toxicity and it is not clear how far it will go.

We are aware that the phase II trial on the association between vemurafenib and bevacizumab was stopped. This trial was mentioned in the Discussion of the old version and now this sentence has been deleted.

One last point, and that is that I think the authors have distorted the supposed value of vessel normalization as a mechanism to increase therapy efficacy. There are increasing numbers of studies coming out, both preclinical and clinical, that an antiangiogenic drug actually diminishes the intratumoral delivery of concurrently administered drugs including chemotherapy or other antibodies such as trastuzumab. Thus, I think the authors should be more careful in their assessment of the supposed benefits of vessel normalization as a clinically relevant mechanism for increasing cancer therapy efficacy. This concern is also mentioned because, once again, all of the mechanism results were undertaken on transplanted subcutaneously grown primary tumors, and I am extremely dubious whether the results would pertain in a general way to melanomas growing in metastatic sites such as the lungs, liver, or brain.

Experimental models demonstrated that the altered vasculature in tumours can be remodeled using low doses of anti-angiogenic agents. This strategy, defined as vascular normalization, can improve tumour perfusion and thereby reduce interstitial pressure, increase oxygen and drug delivery, and ultimately enhance treatment efficacy. This effect was also demonstrated in preclinical models of melanoma (doi: 10.1002/lary.26215; doi.org/10.1016/j.ccr.2014.06.025). The translation of this pre-clinical evidence to the clinics is not easy because it is necessary to find out the best temporal window and the more appropriate drug schedule. However, several phase I and Phase II clinical trials have demonstrated that this process can occur also in some human tumors such as brain tumors, breast and rectal cancers (doi: 10.1073/pnas.1518808112, doi: 10.1073/pnas.1424024112, doi: 10.1093/neuonc/nov085; doi: 10.1158/0008-5472.CAN-11-2464, Nat Med. 2004;10,145-147), while conflicting results have been reported for lung cancer (doi: 10.1073/pnas.1424024112; Cancer Cell 21, 82-91, 2012). We believe that a discussion of a pre-clinical study can propose to the readers the hypothesis that the vascular effect of COMBO can cooperate with its effect on myeloid compartment.

Referee #3 (Remarks):

This manuscript provides evidence that combination of bevacizumab and the BRAF inhibitor PLX4720 is an effective anti-cancer strategy. In fact the study shows that the combination has surprising therapeutic efficacy in preclinical models of melanoma and colon cancer. The study builds on prior work from the group on Braf inhibition and its effect on tumor vasculature. Further, there are a number of intriguing aspects to the study that elevate the work beyond the typical combinatorial study. The authors detail surprising changes in hypoxia, ECM remodeling and fibroblast activation after combination of bevacizumab with PLX4720. A strong point of the study is the transcriptional analysis the authors perform on tumors from mice treated with single agent and combination therapy. Overall, given the fact that the data seem to run counter to what would be expected with an anti-angiogenic agent in combination with a kinase inhibitor, the burden of evidence is high for the conclusions drawn. Thus while the results are intriguing there are a number of challenges that should be addressed. As presented the conclusions appear to be the result of over interpretation of the data presented.

We thank this referee for his/her encouraging and largely positive comments.

Major concerns:

1. It is not surprising that blocking VEGF and BRAF activity is more effective in controlling tumor growth but the changes in tumor perfusion and hypoxia after combination therapy are surprising. I realize this was reported in the PNAS 2012 paper but in that paper single agent therapy was used and the mechanism of how PLX4720 alters perfusion and hypoxia is not clear. The PNAS 2012

paper states that PLX4720 does not directly affect endothelial cells but reduces proangiogenic molecule expression by tumor cells. However, the authors should look at endothelial cell proliferation in the tumors treated with PLX4720 and combo; this could be done by IHC for a proliferation marker and an endothelial cell marker. This would help clarify how PLX4720 increases perfusion and reduces hypoxia.

According to the suggestion of this referee, we stained sections of A375 tumors with an Ab-anti Ki67 and anti-CD31 to study the effect of COMBO and PLX4720 on EC proliferation. As shown in the below figure, we did not appreciate any significant effect on EC proliferation induced by PLX4720 alone or combined with bevacizumab.

Figure 1. Representative images of EC proliferation determined by CD31 (magenta) and Ki67 (green) immunofluorescence staining in A375 xenografts treated as indicated for 2 weeks. DAPI in blue.

Because the lack of any evident combinatorial effects between bevacizumab and PLX4720 on tumor vessels (see also Figure 2), we believe that experiments to dissect the PLX4720 mechanism on vasculature are important but out the scope of the present work

Is this surprising effect of combination therapy limited to BRAF mutant tumors? There are numerous studies in multiple tumor models showing that anti-angiogenic therapy in combination with another agent increases hypoxia and reduces perfusion, thus the data presented here are quite distinct and suggest that PLX4720 has activity beyond tumor cells.

I completely agree with the suggestion of this referee that PLX4720 “has activity beyond tumor cells”. This is supported by our analysis of human and murine transcriptome in A375 xenograft (see Figure 3) and by data of the literature that show the activity BRAF inhibitors. In particular Hirata E (Cancer Cell 27, 574, 2015; quoted in this paper) showed that PLX4720 up-regulates the expression PDGF receptor of melanoma fibroblasts and stimulates their ability to contract collagen. This point has been commented in the Discussion (see page 22, lines 11-15)

If the effect is solely due to a reduction in secretion of factors from tumor cells, which factors are most critical to the stromal changes? Can the authors speculate and provide some protein data to validate that speculation. They do not need to functionally validate the candidate soluble factors.

According to this appropriate criticism we validated the prediction of IPA analysis shown in the old version of the manuscript and now reported in Fig 6D indicating that TGFβ was a strong candidate. Figures 6E and F shows that PLX4720 as well as bevacizumab reduce human TGFβ1 expression, while COMBO completely abrogated the expression of this soluble mediator. On the other hand PLX4720 and COMBO do not modify the expression of murine Tgfb1 transcript. (see page 15 lines 20-23 and page 16 line 1-2)

Acute therapy with an anti-VEGF can result in transient vascular normalization. However there are a few caveats. First bevacizumab only inhibits human VEGF, thus in the xenograft setting mouse VEGF secreted from the stromal cells in the tumor microenvironment is still functional. The authors should evaluate the level of mouse VEGF protein, is it possible that PLX4720 reduces mouse VEGF production/secretion?

By blot analysis, we investigated the amount of murine VEGF in A375 xenografts in mice treated for 2 weeks with the indicated treatments. As shown in the below figure we did not observe any significant modifications of murine VEGF in xenografts differently treated.

Figure 2. Western blot of mouse VEGF in total protein extract from A375 tumors treated as indicate. The following sentence has been added at page 8 (lines 3-6)

Furthermore, the lack of any evident combinatorial effects between bevacizumab and PLX4720 on tumor vessels, and on the amount of murine VEGF detected in the xenografts (not shown), suggests that the enhanced antitumor activity observed with COMBO is likely independent of angiogenesis

Chronic therapy should be explored. The data presented in Figure 7 speak to this somewhat. What happens to vascular density in the responding and relapsing tumors?

We analysed MVA and hypoxia in responding and relapsing tumors. Figure S6 shows that there are not differences in MVA but hypoxia was absent in responders and appeared in relapsing tumors.

2. The other surprising changes observed in tumors treated with combination therapy are the dramatic changes in immune cell recruitment and fibroblast activation and the change in collagen deposition.

a. Again the question of the effect of PX4720 on stromal cells is relevant. Does PLX4720 directly affect immune cells or fibroblasts or are the changes in the tumor microenvironment due to the alleviation of hypoxia? This should be addressed with at minimum in vitro studies evaluating the effect of PLX4720 on a fibroblast cell line and the discussion should be expanded in this area.

see above

The authors should directly evaluate TGF β protein as this growth factor is often prevalent in solid tumors and relevant to the phenotypes they are reporting (e.g., fibroblast, immune cells, and collagen deposition).

see above

3. A therapy study with the same treatment groups in a non-BRAF mutant tumor should be provided.

We respectfully disagree with the reviewer that the requested experiment is unnecessary. While PLX4720 (an analogue of vemurafenib) has known antiproliferative activity in BRAF mutant cells, it also induces a paradox effect on BRAF wild-type and RAS mutant tumors, leading to hyperproliferation (10.1038/nature08833). Indeed, we note that patients with BRAF wild-type tumors are rightly excluded from treatment with BRAF inhibitors, therefore the experiment proposed by the reviewer would have no translational or clinical relevance. We believe that the specificity of PLX4720 against mutant BRAF is so well established that a control experiment in wild type cells would not be necessary, and we further note that the use of animals for the proposed experiment would not fully comply to the 3R principles.

4. Can the authors speculate on potential biomarkers that could be evaluated in patients treated with a BRAF inhibitor that would predict response or justification for combination with another therapy? *We thank the reviewer for raising this interesting point. Despite intense research in the field, there are no established molecular biomarkers of response to antiangiogenic therapy. On the other hand, the mutational status of BRAF is an approved predictive biomarker of response to BRAF inhibitors.*

Yet, it is known that not all BRAF mutant melanoma patients respond to BRAF inhibition, and even when patients respond, responses are transient. Given the current dismal scenario, we speculate that the combination of BRAF inhibitors with antiangiogenic therapy should be tested in BRAF mutant melanomas in carefully designed pilot clinical studies that, in addition to refined imaging methods (MRI, PET), would also encompass a strong translational component aimed at collecting on-treatment biopsies as well as plasma samples. The clinical development strategy may also include a triplet combination with immune checkpoint modulators, as briefly mentioned in the discussion.

5. In general the number of animals in each experiment should be clarified. Further it would be useful to the reader to report the tumor volume of the tumors being compared by IHC or functional analysis (e.g., perfusion/hypoxia). This is especially relevant for the responding v relapsing tumors. *We carefully checked all legend of the figures and included the requested information on the number of mice. As concern the second issue, all figures are representative of a tumor with a size included the in median size of the group analyzed. This point was specified at page 24, line 18-19. Furthermore I underline that we quantified almost all immunofluorescence pictures and the bar graphs represent the mean of all tumors analysed (page 24, lines 8-11).*

Minor concerns Why is CD146 used for evaluation of the vascular compartment in Colo205 xenografts. The images with CD146 look quite distinct from the CD31 used for MVD studies in the melanoma sections. It is hard to compare the effect between the two xenografts when different vascular markers are used.

The staining of vessels in COLO205 tumors has now been performed with Ab-anti CD31 (see Figure S2A)

2. The images used for PLX4720 and Bev in panel 6E are the same
We apologize for this mistake and we substituted the figure 6A

3. F4/80 is misspelled -line 5 page 12
This typo has been corrected

3rd Editorial Decision

26 October 2016

Thank you for the submission of your revised manuscript to EMBO Molecular Medicine. We have now received the enclosed reports from the Reviewers who were asked to re-assess it.

We are sorry that it has taken longer than usual to get back to you on your manuscript. We experienced some difficulties in securing the evaluations in a timely fashion. Further to this, our decision required further consultation with the reviewers followed by extensive internal discussion

As you will see, while reviewer 3 is now satisfied with the manuscript, , reviewer 2, still has significant concerns on the manuscript. Reviewer 1 instead, while also supportive, has a few remaining issues.

Reviewer 2 while appreciating the study, remains unconvinced that there is a sufficiently important message contained within the manuscript. For instance s/he does not think that a BRAF inhibitor will be used with bevacizumab in the clinic for the treatment of melanoma.

After reviewer cross-commenting and internal discussion, we agreed that the doubts on the therapy strategy and on the limitations due to the short treatment are reasonable ones. However, we concluded that these considerations should not impede publication given the interesting biology and overall implications of the data. All in all the data are strongly indicative of a reprogramming of the tumor microenvironment that favors immune activation.

In conclusion, I must ask you to carefully consider reviewer 2's points and incorporate further cautionary elements of discussion. Please also address reviewer 1's concern over the quality of images and the suggestion to further discuss on immune implications. I am prepared to evaluate your next, final version of your manuscript at the editorial level, provided the concerns are fully addressed including with a rebuttal.

Please submit your revised manuscript within two weeks. I look forward to seeing a revised final

form of your manuscript as soon as possible.

***** Reviewer's comments *****

Referee #1 (Remarks):

The manuscript by Comunanza et al has been significantly improved by the major revisions added. Some irrelevant pieces of information have been removed, while relevant analyses have been incorporated and correct interpretations have been added to the results and discussion sections of the manuscript.

The quality of F4/80 staining appears inferior, presentation of the higher magnification images may solve this problem. However the differences appear convincing and this does not reflect on overall conclusions of the manuscript.

There is, however, an important point. However, that should have been made. The fact that the combination only reduces the size of micrometastases, rather than the number of metastatic colonies suggests the involvement of adaptive immune response rather than direct eradication of circulating and extravasating cancer cells by M1 macrophages. The induction of adaptive immune response may not be direct, but promoted indirectly by the M1 macrophages. This conclusion is underscored by the fact that the number of potential antigen-presenting dendritic cells is dramatically increased. Although the analysis of T-cell infiltration, proliferation and the expression of PD-L1 by the macrophages are beyond the scope of the study, these considerations would significantly improve the discussion.

Referee #2 (Remarks):

I'm writing you about the review of the revised manuscript by Bussolino and his colleagues ("VEGF blockade enhances the anti-tumor effect of BRAFV600E inhibition"). It was interesting to note the reviews by the other two referees of this manuscript which certainly had a more negative than positive tone - even reviewer #3. Based on the sum total of criticisms - conceptual, technical, and methodological - along with the responses of the authors, and the revisions made, my original opinion remains unchanged. I don't think this manuscript meets the high standards that are expected for publication in EMBO Mol Med. Here I will briefly confine my comments to the responses about my particular concerns. I still feel that the models used are deficient in terms of clinical translation and by that I mean that the predominant use of short term therapies involving treatment of subcutaneous tumors. There has been some improvement by the inclusion of limited amount of data evaluating the response of metastases established after intravenous inoculation of tumor cells. However, as also pointed out by one of the other reviewers, the lack of any data in immune competent mice also reduces the rigor of the experimental approach. Finally, as mentioned, I do not foresee this particular combination approach having any future in the clinic. If VEGF blockade will have some future promise as a clinical treatment strategy for melanoma, it will in all likelihood involve combination with immune checkpoint inhibitor immunotherapy. This is because of increasing evidence that VEGF blockade can impact the efficacy of immune checkpoint inhibitors. Indeed, the work of Steve Hodi and colleagues evaluating combination of bevacizumab and ipilimumab in melanoma patients highlight this point ("Bevacizumab plus ipilimumab in patients with metastatic melanoma" *Cancer Immunol Res.* 2014 Jul;2(7):632-42). Indeed, there is growing evidence that VEGF can act as a local immunosuppressive factor in tumors - another reason to have undertaken some therapy experiments in immune competent mice.

While I believe there is certainly some interesting results in the manuscript and that it deserves publication, my feeling is that a cancer research journal such as *Mol Cancer Therapeutics* or the *Int J Cancer* would be more suitable for publication of this kind of study.

Referee #3 (Comments on Novelty/Model System):

The revision has produced a high quality study that provides novel analysis of why anti-VEGF + a

BRAF inhibitor can be effective preclinically. The paper provides insight that can be exploited in the clinical testing of such a combination, which has been attempted unsuccessfully in the clinic.

Referee #3 (Remarks):

The authors have done a good job at addressing the major concerns of the initial review. The paper has been substantially revised and the revision has much more impact and depth. The additional studies on the immune cell component and ECM changes in the combo treated tumors are considerable strengths. The study is now much more than a combination therapy exercise. There are still open questions and they did not address everything directly but the impact of the study is substantial for basic cancer biology and cancer therapy.

3rd Revision - authors' response

08 November 2016

I thank the referees for their useful suggestions and criticisms and point-to-point reply is here provided.

Referee #1

I thank this referee for his/her positive comments. The following points have been further addressed:

1. *"The quality of F4/80 staining appears inferior..."*
To overcome this point and maintain a magnification homogeneity between the different pictures we replaced F4/80 staining in the Figure 4B with higher magnification (40 X) images.
2. *"There is, however, an important point that should have been made. The fact that the combination only reduces the size of micrometastases, rather than the number of metastatic colonies suggests the involvement of adaptive immune response rather than direct eradication of circulating and extravasating cancer cells by M1 macrophages. The induction of adaptive immune response may not be direct, but promoted indirectly by the M1 macrophages. This conclusion is underscored by fact that the number of potential antigen-presenting dendritic cells is dramatically increased. Although the analysis T-cell infiltration, proliferation and the expression of PD-L1 by the macrophages are beyond the scope of the study, these considerations would significantly improve the discussion."*
I completely agree with this comment and we modified two paragraphs of the Discussion (old version, page 21, lines:1-25; page 22, lines:1-4; now page 21, lines 525; page 22 lines 1-5).

Referee #2

I thank this reviewer for his/her work. Obviously I disagree with the statement that the other comments of referees 1 and 2 were "more negative than positive" and in particular referee #3 states that our data can be "exploited in the clinical testing of such a combination, which has been attempted unsuccessfully in the clinic". A second issue raised by this referee is the use of short-term therapies. As reported in my previous rebuttal this strategy is largely based on ethical issues and to respect the 3R rules (replacement, reduction and refinement) recommended in pre-clinical experimentation. Because the life expectancy of the mouse ranged from 2 to 3 years, I underline the fact that 1 month of life in mice corresponds to 14 months in humans (L. Demetrius Ann N.Y. Acad.Sci. 1067:66-82, 2006). Accordingly, the criticisms about our schedule treatment are not completely appropriate. As far as concerns the use of immune-compromised mice, I discussed this point in the previous rebuttal by clearly claiming also in the manuscript the limit of our approach that represents a balanced compromise largely exploited in pre-clinical experiments to model aspects of human oncogenesis.

Finally, I want to emphasize that the paper by Dr Hodi (Cancer Immunol Res. 2014 Jul;2(7):632-42) on the combination between bevacizumab and ipilimumab mentioned by this referee was quoted and discussed in the old version of this manuscript.

Because the principal aim of our study was just to understand the molecular and cellular tumor modification resulting by the combination of two drugs largely used in clinical oncology, we deleted in the text body all kind of clinical perspective. In particular the last paragraph of old version (page

22, line 16-22) was changed as follows:

Overall, our data suggest that the removal of VEGF can increase the efficacy of BRAF^{V600E} inhibitors by inducing vascular normalization, overcoming immune tolerance and modifying the role played by CAFs. Further experiments will be required to investigate how T and B cells, which are absent in CD1 mice used in this work, modify the functional performance of the association between BRAFV600E targeting and VEGF removal.

Similarly, the last sentence of the abstract of the old version (page 2, lines 12-14) was changed as follows:

Collectively, our findings offer new biological rationales for the management of clinical resistance to BRAF inhibitors based on the combination between BRAF^{V600E} inhibitors with anti-angiogenic regimens.

Referee #3

I thank this reviewer for the acceptance of the experiments provided in the revised version of the Manuscript done accordingly to his/her comments.

Corresponding Author Name: Federico Bussolino

Manuscript Number: EMM-2015-05774-V3